


# A monthly tidal envelope classification approach for semi-diurnal
# regimes with variability in $S_2$ and $N_2$ tidal amplitude ratios
Do-Seong Byun[1], Deirdre E. Hart[2]
[1]Ocean Research Division, Korea Hydrographic and Oceanographic Agency, Busan 49111, Republic of Korea
[2]School of Earth and Environment, University of Canterbury, Christchurch 8140, Aotearoa New Zealand
*Correspondence to*: Deirdre E. Hart (deirdre.hart@canterbury.ac.nz)
**Abstract.** In a world of increasing coastal inundation hazards, an understanding of daily through to monthly tidal envelope
characteristics is fundamental to resilient coastal management and development practices. For decades, scientists have
described and compared daily tidal forms around the world's coasts based on the four main tidal amplitudes. Our paper builds
on this 'daily' method by adjusting the constituent analysis to distinguish the different monthly types of tidal envelope
occurring in the semi-diurnal coastal waters around Aotearoa New Zealand. Analyses of tidal records from 23 stations are
used, alongside data from the FES2014 tide model database and theoretical experiments, in order to find the key characteristics
and constituent ratios of tides that can be used to classify monthly tidal envelopes. The resulting monthly tidal envelope
classification approach described ($F_M^S$) is simple, complementary to the successful and much used daily tidal form factor (F),
and of use for coastal flooding, climate change and maritime operation management and planning applications in semi-diurnal
regimes.
**1 Introduction**
Successful human-coast interactions in the world's low-lying areas are predicated upon understanding the temporal and spatial
variability of sea levels (Cartwright, 1999; D'Onofrio et al., 1999; Nicholls et al., 2007). This is particularly the case in island
nations like Aotearoa New Zealand (ANZ), where over 70% of the population reside in coastal settlements (Stephens 2015).
An understanding of tidal water level variations is fundamental to resilient inundation management and coastal development
practices in such places (Masselink et al., 2014; Olson, 2012; Pugh, 1996), as well as to accurately resolving non-tidal signals
of global interest, such as in studies of sea level change and gravimetry (Egbert et al., 1994; Stammer et al., 2014).
In terms of daily cycles, tidal form factors or form numbers ($F$) based on the amplitudes of the four main tidal constituents
($K_1$, $O_1$, $M_2$, $S_2$) have been successfully used to classify tidal observations from the world's coasts into four types of tidal
regime for nearly a century (Figure 1 a). Originally developed by van der Stok (1897), with a fourth category added by Courtier
(1938), these simple and useful form factors comprise the ratio between the combined $K_1$ and $O_1$ diurnal amplitudes versus
the combined $M_2$ and $S_2$ semi-diurnal amplitudes (Table 1). The resulting form factors classify tidal regimes into those which
roughly experience one high and one low tide per day (diurnal regimes); or two approximately equivalent high and low tides
per day (semidiurnal regimes); or two unequal high and low tides per day (mixed semidiurnal dominant or mixed diurnal
dominant regimes) (e.g. Defant 1958).
Albeit not part of their original design, some interpretation of the tidal envelope types observed at fortnightly and monthly
timescales has accompanied use of daily tidal form classifications (e.g. Pugh, 1996; Pugh & Woodworth, 2014). Whereas the
daily tidal form factor identifies the number and form (equal or mixed) of tidal height cycles typical within a lunar day (i.e. 24
hours and 48 minutes) at a particular site, a tidal envelope describes the maximum and minimum boundaries of tidal height
cycles occurring across a specified timescale at that site. The envelope timescale of interest in this paper is monthly.





Tidal envelopes at monthly scales depend on tidal regime. In general, semi-diurnal tidal regimes often feature two spring-neap
tidal cycles per synodic (lunar) month (Table 1). These two spring-neap tidal cycles are usually of unequal magnitude, due to
the effect of the moon's perigee and apogee, which cycle over the period of the anomalistic month. In contrast, diurnal tidal
regimes exhibit two pseudo spring-neap tides per sidereal month. For semi-diurnal regions where the $N_2$ constituent contributes
significantly to tidal ranges, tidal envelope classification should consider relationships between the $M_2$, $S_2$, and $N_2$ amplitudes.
The waters around ANZ represent one such region: here the daily tidal form is consistently semi-diurnal, but large differences
occur between sites within this region in terms of their typical tidal envelope types over fortnightly to monthly timescales.
The primacy placed on the four main amplitudes used in daily tidal form calculations has influenced the constituents examined
in comparisons between global tide models and satellite altimeter data (e.g. Andersen, 1995; Stammer et al., 2014),
emphasizing the importance of daily and spring-neap constituents. Far less attention has been paid to of the importance of
other constituents in modern tidal research. More than eighty years after the development of the ever-useful daily tidal form
factors, attention to the regional distinction between different tidal envelope types within the semi-diurnal category is also
needed, and forms the motivation for this paper. In this first explicit attempt to classify monthly tidal envelope types, we
examined the waters around ANZ, a strong semi-diurnal regime with relatively weak diurnal tides (daily form factor $F < 0.15$)
and variation in the importance of the $S_2$ and $N_2$ amplitude ratios. The result is an approach for classifying monthly tidal
envelope types that is transferable to any semi-diurnal regime. As well as providing greater understanding of the tidal regimes
of ANZ, we hope that our paper opens the door for new international interest in classifying tidal envelope variability at multiple
timescales, work which would have direct coastal and maritime management application including contributing to explanations
of the processes behind delta city coastal flooding hazards and their regional spatial variability.
**2 Methodology**
**2.1 Study area**
Aotearoa New Zealand is a long (1600 km), narrow (≤400 km) country situated in the south-western Pacific Ocean and
straddling the boundary between the Indo-Australian and Pacific plates. Its three main islands, Te Ika-a-Māui or the North
Island, Te Wai Pounamu or the South Island, and Rakiura or Stewart Island, span a latitudinal gradient between about 34° and
47° South. The tidal regimes in the surrounding coastal waters are semi-diurnal, with variable diurnal inequalities, and absolute
tides that span micro through to macro tidal ranges. Classic spring-neap cycles are present in western areas of ANZ, while
eastern areas feature distinct perigean-apogean influences (Byun and Hart, 2015; Heath, 1977, 1985; LINZ, 2017b; Walters et
al., 2001).
Highly complex tidal propagation patterns occur around ANZ, including a complete semi-diurnal tide rotation: contrary to the
southern hemisphere Coriolis Effect, the tide generally circulates around this country in an anti-clockwise direction. This
occurs due to the forcing of $M_2$ and $N_2$ tides by two amphidromes, situated northwest and southeast of the country, producing
trapped Kelvin waves; while the $S_2$ and $K_1$ tides exhibit a single wave front generated by an amphidrome to the southeast, plus
refraction of a trapped wave around the South Island. Around Te Moana-o-Raukawa or Cook Strait, the waterway between the
two main islands, tides travelling north along the east coast run parallel to tides travelling south along the west coast. The
pronounced differences between these east/west tidal states, combined with their tidal range differences, together produce
marked differences in amplitude and strong current flows through Cook Strait (Heath, 1985; Walters et al., 2001, 2010).
**2.2 Data analysis approach**
Year-long sea level records were sourced from a total of 23 stations spread around ANZ (Figure 2): eighteen 1 minute-interval
records from Land Information New Zealand (LINZ, 2017a); and five 1 hour-interval records from the National Institute of
Water and Atmospheric Research (NIWA, 2017). For both the LINZ and NIWA data, years with good quality hourly data



were selected for analysis from amongst multi-year records. The 23 tidal records were harmonically analyzed using T_Tide
(Pawlowicz et al., 2002) to examine spatial variation in the main tidal constituents' amplitudes, phase-lags, and amplitude
ratios, between regions and in comparison with their tidal potential values from Equilibrium Theory (see Table A1 for raw
results). An additional set of tidal constituent amplitude data was sourced from Tables 1 and 3 of Walters et al. (2010), derived
from 33 records of between 14 and 1900 days length, from around the greater Cook Strait area between ANZ's two main
islands, where spring-neap tides reach the strongest in the country.
We then classified the monthly tidal envelope types found around ANZ based on detailed examination of constituent ratios
produced from the tidal harmonic analysis results, as well as data from the FES2014 tide model (see Carrere et al., 2016 for a
full description of this database), and experimental plots of the different tidal envelope types generated from this constituent
data. Due to the strong semi-diurnal tidal regimes in the study area, and similar to the approach of Walters et al. (2010), we
were able to ignore sidereal ($K_1$, $O_1$) effects and simply consider the effects of spring-neap ($M_2$, $S_2$) and perigean-apogean
cycles ($M_2$, $N_2$) in our monthly tidal envelope type characterization.
**3 Results**
**3.1 Key tidal constituent amplitudes and amplitude ratios**
In order to better understand the key constituents responsible for shaping tidal height forms around ANZ, we first mapped
variability in the amplitudes of the semi-diurnal and diurnal constituents listed in Table 1 (Figure 3) and of the ratio values of
the semi-diurnal constituent amplitudes (Figure 4). Table 2 summarizes these data, and contrasts them with those from Defant's
(1958) Equilibrium Theory, while Table A1 catalogues the detailed data results.
Tidal amplitude ratio comparisons confirmed that the waters around ANZ are dominated by the three astronomical semi-
diurnal tides: $M_2$, $S_2$ and $N_2$ (Table 2), the combination of which can generate fortnightly spring-neap tides ($M_2$ and $S_2$) and
monthly perigean-apogean tides ($M_2$ and $N_2$). Figure 3 reinforces the relatively minor magnitudes of diurnal constituent
amplitudes ($O_1$, $K_1$), as well as revealing the stronger west coast amplitudes of the spring-neap cycle generating constituents
($M_2$ and $S_2$), the relatively weak $S_2$ amplitudes overall (half that of Equilibrium Theory),  and the more concentric pattern
around ANZ of the perigean-apogean cycle generating $N_2$ amplitude.
In terms of the semi-diurnal constituent amplitude ratios, Figure 4 and Table 2 show that $\frac{a_{S_2}}{a_{M_2}}$ values cover a broad range around
ANZ (0.04 to 0.47), with most sites exhibiting relatively smaller values (<0.27 at 22 out of 23 sites) than that of Equilibrium
Theory (0.466). In contrast, $\frac{a_{N_2}}{a_{M_2}}$ ratios were found to be more stable around ANZ (values ranging from 0.16 to 0.23) and similar
in magnitude to Equilibrium Theory (i.e. 0.191). By grouping the constituent amplitude and amplitude ratio results (Figures 3
to 4), we were able to distinguish four distinct monthly tidal envelope regimes around ANZ (Table 2).
•   Firstly, '*spring-neap*' type tidal regimes occur where the $S_2$ tide amplitude is large compared to that of the $N_2$ (Table
2, Figure 3). In these areas there are two spring-neap tides per month with similar ranges, and negligible influence of
perigean-apogean cycles. Such a regime occurs in the Kapiti and Cook Strait area (Figure 1), where the $N_2$ and $M_2$
amplitudes reduce by 75 to 90%, but the $S_2$ amplitude reduces by only about 30%, compared to on adjacent coasts.
•   In direct contrast, there are '*perigean-apogean*' type tidal regimes, in areas where the $N_2$ amplitude strongly
dominates over the $S_2$ (Table 2, Figure 3). In this type of tidal regime, the $M_2$ and the $N_2$ tides combine to produce
strong signals over anomalistic timeframes (27.5546 days). Hence the highest tidal ranges in any given month occur
in relation to the perigee, when the moon's orbit brings it close to Earth, rather than in line with the moon's phase, as
is typical in spring-neap regimes. This type of regime occurs, for example, around the northern Chatham Rise near
Kaikoura, and as far north as Castle Point on the east coast of the South Island.



The remaining coastal waters around ANZ can be separated into two tidal sub-regions, one with strong spring-neap signals
and the other with strong perigean-apogean signals, but both with overall mixed or *intermediate* monthly tidal envelope types
(Table 2). We distinguished these two envelope types via the combined variability of the ratios of $\frac{a_{S_2}}{a_{M_2}}$ and $\frac{a_{N_2}}{a_{M_2}}$ (i.e. of the
spring-neap cycle; and perigean-apogean cycle forming tides, respectively). By examining these ratios we take account of the
moderating influence of the $M_2$ tide at both synodic and anomalistic timeframes. In brief, the $\frac{a_{S_2}}{a_{M_2}}$ and $\frac{a_{S_2}}{a_{N_2}}$ ratios vary widely
around ANZ, with highest values in the west, lowest values in the east, and intermediate values to the north and south (Figure
4). By comparison, $\frac{a_{N_2}}{a_{M_2}}$ values are relatively stable and high, except in a relatively small area of central Cook Strait, where this
ratio drops and thus spring-neap cycles predominate (see 'spring-neap' type regimes above). The combined variability in these
two ratios means that, except where we find 'perigean-apogean' or 'spring-neap' type monthly tidal envelope types, spring-
neap tides do occur but the overall monthly envelope shape is fundamentally altered (asymmetrically) due to the perigean-
apogean influence.
• In the first of the 'intermediate' monthly envelope sub-regions, tides exhibit two dominant, but unequal, spring-neap
cycles per month due to a subordinate, but still influential, perigean-apogean effect. We term this type of monthly
tidal envelope an '*intermediate, predominantly spring-neap*' type regime. Here values of $\frac{a_{S_2}}{a_{N_2}}$ are ≥1, with $S_2$
amplitudes reaching only around 17 to 50% those of the $M_2$ constituent (Figures 3 to 4; Table 2). Also in these areas,
values of $\frac{a_{S_2}+a_{N_2}}{a_{M_2}}$ are ≥0.45. This type of tide occurs, for example, at the Westport and Puysegur sites.
• In the other 'intermediate' monthly envelope sub-region, tides exhibit a mainly perigean-apogean form with a weaker,
but noticeable, spring-neap signal: we term this envelope type as '*intermediate, predominantly perigean-apogean*'.
Here values of $\frac{a_{S_2}}{a_{N_2}}$ sit between 0.3 and <1, while values of $\frac{a_{S_2}+a_{N_2}}{a_{M_2}}$ are 0.3 to 0.4 (Figure 4, Table 2). This type of
tide occurs, for example, at the Auckland and Sumner sites.
Figure 5 illustrates the four types of monthly tidal envelope found around ANZ as idealized types, two with stronger spring-
neap signals (hereafter referred to as Types 1 and 2, see Figure 5 a-b) and two with stronger fortnightly perigean-apogean
signals (hereafter Types 3 and 4, see Figure 5 c-d).

### 3.2 A monthly tidal envelope factor ($F_M^S$) for semi-diurnal regimes

The four types of monthly tidal envelope types found around ANZ are essentially different combinations of spring-neap and
perigean-apogean signals. Thus, in a similar manner to van der Stok's (1897) method for calculating *daily* tidal form factors,
a *monthly* tidal envelope factor ($F_M^S$) may be calculated for semi-diurnal tidal regions, including that of ANZ, according to:
$F_M^S = \frac{a_{M_2}+a_{N_2}}{a_{M_2}+a_{S_2}},$      (1)
which can be further expressed as:
$F_M^S = \frac{1+\frac{a_{S_2}}{a_{M_2}}x}{1+\frac{a_{S_2}}{a_{M_2}}},$ with $x = \frac{a_{N_2}}{a_{S_2}}$      (1a)
for areas characterized by more stable (e.g., lower variability) values of $\frac{a_{S_2}}{a_{M_2}}$ compared to $\frac{a_{N_2}}{a_{S_2}}$, or as:
$F_M^S = \frac{1+\frac{a_{N_2}}{a_{M_2}}}{1+\frac{a_{N_2}}{a_{M_2}}y},$ with $y = \frac{a_{S_2}}{a_{N_2}}$      (1b)
for areas characterized by more stable (e.g., lower variability) values of $\frac{a_{N_2}}{a_{M_2}}$ compared to $\frac{a_{S_2}}{a_{N_2}}$.





$F_M^S$ takes into account the roles of the $S_2$ and $N_2$ tides in spring-neap and perigean-apogean cycles, while also factoring in the
strong $M_2$ tide influence in both types of cycle. $F_M^S$ may be used to classify the monthly tidal envelope types of any semi-
diurnal region (i.e. where $F<0.25$) based on the analysis of constituent amplitudes and ratios from local data . Below we explain
the four steps undertaken to successfully set the boundaries between different monthly tidal envelope types, thereby classifying
the region's tides, using our ANZ case study data.

**Step 1: Separating regimes dominated by spring-neap versus perigean-apogean signals**

Fundamentally, in any semi-diurnal tidal regime ($F<0.25$) anywhere in the world where $\frac{a_{N_2}}{a_{S_2}} < 1$, spring-neap cycles will be a
clear feature of the tidal height records (Table 1). This applies to the waters around ANZ. Thus, we set an initial boundary
between different monthly tidal envelope types at $\frac{a_{N_2}}{a_{S_2}} = 1$ (Table 4). That is, regimes where $\frac{a_{N_2}}{a_{S_2}} < 1$ feature stronger spring-
neap cycles, while regimes with $\frac{a_{N_2}}{a_{S_2}} > 1$ feature stronger perigean-apogean signals. As summarized in Table 2, compared to
areas that experience stronger spring-neap influences, areas of the ANZ coast with stronger perigean-apogean influences are
characterized by relatively smaller $S_2$ amplitudes (2-18 cm), with stronger $N_2$ amplitudes (10-22 cm).

**Step 2: Separating regimes with consistent versus irregular and unequal spring-neap**

Tidal regimes with stronger spring-neap signals (i.e. where $\frac{a_{N_2}}{a_{S_2}} <1$) include places where spring-neap cycles occur as
consecutive fortnightly cycles of similar magnitude (hereafter labelled Type 1 or 'spring-neap' type regimes), and places where
spring-neap signals dominate but with noticeable variability in the magnitudes of consecutive cycles due to subordinate
perigean-apogean influences (hereafter labelled Type 2 or 'intermediate, spring-neap' regimes). In ANZ the strongest spring-
neap influence occurs in the greater Cook Strait area, including at Kapiti where harmonic analysis revealed $\frac{a_{N_2}}{a_{S_2}} = 0.35$ (Table
A1).
To set a boundary between Types 1 and 2 in any semi-diurnal tidal regimes around the world (and between Types 3 and 4 as
explained below), it was necessary to take account of the moderating influence of the $M_2$ amplitude compared to the
magnitudes of the $S_2$ and $N_2$ amplitudes, since the $M_2$ constituent influences monthly tidal envelopes at both synodic (spring-
neap) and anomalistic (perigean-apogean) timescales (Table 1). In order to do this, experiments were conducted to explore
two additional ratios:
    i.    the ratio of the 'annual' maximum tidal range to the subsequent tidal range (MTR); and
    ii.    the ratio of the 'annual' maximum spring tide range to the subsequent spring tide range (MSR).
We determined that the monthly tidal envelope boundary ($F_M^S$) between spring-neap (Type 1) and intermediate, spring-neap
dominant (Type 2) regimes would occur at point where the MTR and MSR tidal range ratios exhibited the same value. 360
days of synthetic tidal range data were generated under conditions of $F=0.25$ (the boundary between 'semi-diurnal' and 'mixed,
mainly semi-diurnal' type daily forms); and $a_{M_2} = 3a_{S_2}$; $a_{K_1} = a_{O_1}$; and $g_{M_2} = g_{S_1} = g_{K_1} = g_{O_1} = 0°$ (a subset of the
assumptions employed by Courtier, 1938).When $F=0.25$, calculations revealed that the MTR value was 0.795. When the MSR
ratio was set to the same value (0.795), calculations revealed that the $F_M^S$ value was 0.795 (Figure A1). Based on this value, a
review of our observation records revealed that the Kapiti site, with its $F_M^S$ value of 0.79, exhibited the only completely spring-
neap dominated site amongst our ANZ records. Hence we found that the boundary between monthly tidal envelope Types 1
and 2 in ANZ would site somewhere between the tidal regimes of Kapiti and the site with the next strongest spring-neap
influence, Manukau, where $F_M^S = 0.93$.



**Step 3: Separating regimes with 'perigean-apogean' and 'intermediate, perigean-apogean dominated' monthly tidal**
**envelopes**
Amongst our case study sites, areas with the most extreme perigean-apogean signal typically exhibited $\frac{a_{S_2}}{a_{M_2}}$ values of about
0.04 to 0.05 (Table A1). In order to determine the boundary between 'perigean-apogean' and 'intermediate, perigean-apogean
dominant' regimes (i.e. Types 3 and 4), we conducted 31 experiments each generating 360 days of synthetic tidal ranges based
on the fixed condition of $\frac{a_{S_2}}{a_{M_2}} = 0.05$, but with $\frac{a_{N_2}}{a_{S_2}}$ values ranging from 3 to 6 at intervals of 0.01 using Eq. (1a). Examining the
shapes of the resultant monthly tidal envelopes, we were able to set a boundary value between Types 3 and 4 regimes at
$F_M^S = 1.15$ in NZ waters (Table 4, Figure 6).
In summary, Figure 7 illustrates the classification of monthly tidal envelope types in the waters around ANZ using $F_M^S$. We
find the west coast is characterized by Type 2 monthly tidal envelopes, with two unequal spring-neap cycles per month. Type
1 monthly tidal envelopes, with their defined spring-neap tides, are only found in the Cook Strait area. This area's defined
spring-neap tides were explored in detail by Walters et al. (2010) - Figure 6 includes a re-analysis of their data using the $F_M^S$
ratios. In contrast, the central east coast shows Type 4, perigean-apogean tidal envelopes, which is unusual for semi-diurnal
regimes internationally (i.e. Figure 1c). Type 3 or intermediate, perigean-apogean dominated monthly tidal envelopes are found
in the rest of the waters surrounding ANZ.
**4 Discussion and conclusion**
Daily tidal water level variations are a key control on shore ecology; access to marine environments via boat and shipping
infrastructure such as ports, jetties and wharves; drainage links between the ocean and coastal hydrosystems such as lagoons
and estuaries; and the duration and frequency of opportunities to access the intertidal zone for recreation and food harvesting
purposes. Fortnightly and monthly tidal envelope variations, such as those associated with spring-neap and perigean-apogean
cycles, have similar moderating roles on human usage of intertidal and shoreline environments, and additionally these medium
term variations in tide levels are important factors in coastal inundation risks (Menéndez & Woodworth, 2010; Stephens 2015;
Stephens et al., 2014; Wood, 1978, 1986;). High perigean-spring tides, for example, interact with extreme weather events
(including low pressures, strong winds and extreme rainfall) to produce significant coastal inundation in low-lying coastal
settlements such as on deltas (Hart et al., 2015).
In a world of rising sea levels, and coastal inundation hazard cascades (Menéndez and Woodworth, 2010), having common
ways of describing different types of tidal envelope is essential for living safely and productively in coastal cities. This paper
has employed observations from ANZ, FES2014 tidal data, and theoretical experiments, to demonstrate a simple approach to
classifying different monthly tidal envelope types, applicable to semi-diurnal regions anywhere. The result is a widely
applicable monthly tidal envelope factor ($F_M^S$) for classifying semi-diurnal regimes based on the amplitudes and amplitude
ratios of three key constituents operating at synodic anomalistic timescales ($M_2$, $S_2$, and $N_2$).
At a very basic level, in any semi-diurnal tidal regime anywhere in the world where the value of $\frac{a_{N_2}}{a_{S_2}} < 1$, then spring-neap
cycles will be clearly visible in tidal height records, either as consecutive fortnightly cycles of similar magnitude (Type 1), or
as a dominant signal with noticeable variability in the magnitudes of consecutive fortnightly cycles, due to a subordinate
perigean-apogean influence (Type 2). Conversely, in semi-diurnal areas of the world's oceans where $\frac{a_{N_2}}{a_{S_2}} > 1$, then perigean-
apogean cycles will be clearly visible, either as singularly evident monthly cycles (Type 4), or as a dominant influence with
subordinate spring-neap signals (Type 3). As illustrated in Sect. 3.2, quantitatively determining the actual boundaries between
monthly tidal envelope Types 1 versus 2, and Types 3 versus 4 regimes at a local scale involves analysis of observational data,
taking into account the moderating influence of the $M_2$ amplitude compared to the magnitudes of the $S_2$ and $N_2$ amplitudes.



Figure 1b illustrates the division of the semi-diurnal areas of the world's oceans into those where spring-neap cycles are the
main monthly tidal envelope influence versus those where the perigean-apogean signal is paramount, while Figure 1c illustrates
areas of the world's oceans where spring-neap signals are minor compared to 'perigean-apogean' influences in the monthly
tidal envelope. The potentially predictable but relatively lower frequency tidal water level fluctuations such as those in our
perigean-apogean monthly envelope classes are an important cause and moderator of coastal inundation hazards in different
locations around the world (e.g. Wood 1978, 1986; Stephens 2015).
The simple approach to classifying monthly tidal envelope types in semi-diurnal regions demonstrated in this paper
complements the existing, commonly used way of describing daily tidal forms based on the amplitudes of the four key, diurnal
($K_1$, $O_1$) and semi-diurnal ($M_2$, $S_2$) constituents (e.g. Defant 1958). We hope that our work inspires other efforts to study tidal
height variations at timescales greater than daily, work which could draw renewed attention to the fundamental role of tidal
water levels in shaping coastal environments, including in hazards such as low-frequency coastal flooding.
**Data Availability**
The tidal data used in this paper are available from LINZ (2017a; 2017b), NIWA (2017) and Walters et al. (2010). Details of
the    FES2014    tide    model    database    are    found    in    Carrere    et    al.    (2016)    and    via
https://www.aviso.altimetry.fr/en/data/products/auxiliary-products/global-tide-fes.html). Appendix 1 contains the data
produced from analysis of these primary resources in this paper.





Ocean Science Discussions — Open Access — EGU


**Appendix 1**

Table A1. Values for 5 tidal harmonic constants, tidal ranges, form factors ($F$), and monthly tidal envelope factor ($F_M^S$) for a semi-diurnal regime used in classifying tidal envelope forms, for 23 stations around New Zealand, and compared to values derived from Equilibrium Theory

| Station name (record used) | $M_2$ | | $S_2$ | | $N_2$ | | $K_1$ | | $O_1$ | | $F$ | $F_M^S$ | Type |
|---|---|---|---|---|---|---|---|---|---|---|---|---|---|
| | $a_i$ (cm) | $g_i$ (deg.) | $a_i$ (cm) | $g_i$ (deg.) | $a_i$ (cm) | $g_i$ (deg.) | $a_i$ (cm) | $g_i$ (deg.) | $a_i$ (cm) | $g_i$ (deg.) | | | |
| Equilibrium Theory | - | - | - | - | - | - | - | - | - | - | 0.68 | - | - |
| Kapiti (2011) | 55 | 280 | 26 | 336 | 9 | 277 | 2 | 195 | 2 | 18 | 0.05 | 0.790 | 1 |
| Manukau (2011) | 109 | 297 | 29 | 332 | 20 | 287 | 6 | 17 | 1 | 287 | 0.05 | 0.935 | 2 |
| Westport (2015) | 113 | 309 | 29 | 348 | 23 | 287 | 2 | 198 | 3 | 40 | 0.04 | 0.958 | 2 |
| Charleston (2015.7.-2016.1) | 106 | 319 | 27 | 344 | 22 | 304 | 3 | 6 | 3 | 243 | 0.05 | 0.962 | 2 |
| Puysegur Point (2012) | 78 | 350 | 19 | 13 | 17 | 335 | 3 | 316 | 4 | 245 | 0.07 | 0.979 | 2 |
| North Cape (2010) | 80 | 230 | 15 | 279 | 16 | 209 | 8 | 10 | 2 | 351 | 0.11 | 1.011 | 3 |
| Boat Cove, Raoul Island (2012) | 50 | 208 | 9 | 287 | 10 | 176 | 5 | 43 | 3 | 44 | 0.14 | 1.017 | 3 |
| Dog Island (2011) | 91 | 33 | 18 | 57 | 21 | 6 | 2 | 119 | 4 | 60 | 0.06 | 1.028 | 3 |
| Auckland (2011) | 112 | 216 | 17 | 275 | 22 | 192 | 7 | 356 | 2 | 324 | 0.07 | 1.039 | 3 |
| Fishing Rock, Raoul Island (2011) | 52 | 206 | 8 | 283 | 11 | 178 | 5 | 35 | 2 | 41 | 0.12 | 1.050 | 3 |
| Lottin Point (2011) | 70 | 195 | 9 | 262 | 14 | 168 | 6 | 352 | 2 | 328 | 0.1 | 1.063 | 3 |
| Tauranga (2011) | 70 | 211 | 9 | 277 | 14 | 186 | 5 | 360 | 1 | 330 | 0.08 | 1.063 | 3 |
| Korotiti Bay (2011) | 78 | 207 | 11 | 265 | 16 | 181 | 6 | 349 | 1 | 317 | 0.08 | 1.056 | 3 |
| Moturiki (2011) | 73 | 189 | 10 | 265 | 15 | 156 | 5 | 173 | 1 | 136 | 0.07 | 1.060 | 3 |
| Green Island (2011) | 73 | 81 | 9 | 91 | 17 | 50 | 3 | 93 | 4 | 44 | 0.08 | 1.084 | 3 |
| Port Chalmers (2011) | 77 | 112 | 9 | 112 | 17 | 89 | 3 | 270 | 3 | 247 | 0.07 | 1.093 | 3 |
| Sumner (2011) | 84 | 136 | 6 | 151 | 18 | 109 | 5 | 273 | 3 | 245 | 0.09 | 1.133 | 3 |
| Gisborne (2010) | 64 | 176 | 5 | 251 | 14 | 148 | 4 | 336 | 1 | 275 | 0.07 | 1.130 | 3 |
| Napier (2011) | 64 | 167 | 4 | 240 | 14 | 138 | 3 | 298 | 2 | 221 | 0.07 | 1.147 | 3 |
| Kaikoura (2011) | 65 | 146 | 3 | 171 | 14 | 117 | 4 | 275 | 4 | 233 | 0.12 | 1.162 | 4 |
| Owenga, Chatham Island (2011) | 48 | 149 | 2 | 224 | 10 | 119 | 2 | 246 | 2 | 179 | 0.08 | 1.160 | 4 |
| Castlepoint (2011) | 63 | 159 | 3 | 225 | 14 | 129 | 3 | 280 | 3 | 219 | 0.09 | 1.167 | 4 |
| Wellington (2011) | 49 | 148 | 2 | 352 | 11 | 116 | 2 | 268 | 3 | 219 | 0.1 | 1.176 | 4 |
| Overall range | 48-113 | 33-350 | 2-29 | 13-352 | 9-23 | 6-335 | 2-8 | 6-360 | 1-4 | 40-351 | 0.04-0.14 | 0.79-1.176 | 1-4 |



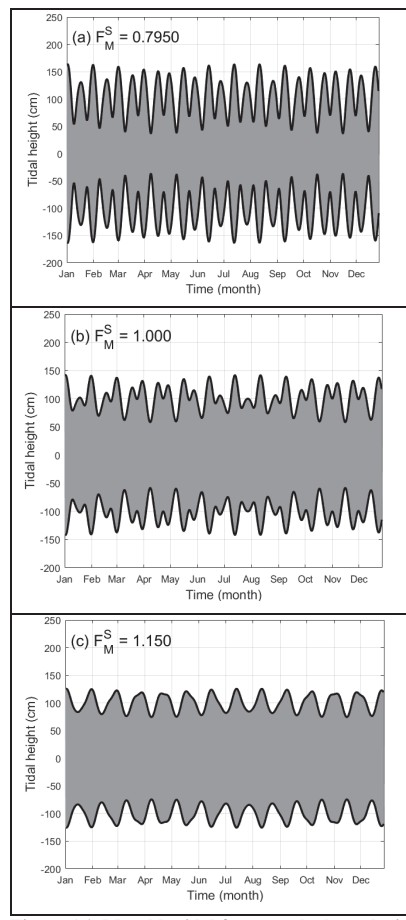

**Figure A1. Monthly tidal forms at the boundaries of the (a) '*spring-neap*' versus '*intermediate, spring-neap dominant*' tidal forms; at the (b) '*intermediate, spring-neap dominant*' versus '*intermediate, perigean-apogean dominant*' tidal forms; and at the (c) '*intermediate, perigean-apogean dominant*' versus '*perigean-apogean*' tidal forms, produced using the conditions summarized in Table A1.**



## Author contribution

Both authors conceived of the idea behind this paper. DH produced the initial manuscript draft. D-SB analyzed the tidal data and wrote the results sections. Both authors worked on and finalized the full manuscript.

## Competing interests

The authors declare that they have no conflict of interest.

## Acknowledgements

We are grateful to Land Information New Zealand (LINZ) and the National Institute of Water and Atmospheric Research (NIWA) for supplying the tidal data used in this research. Thank you to the University of Canterbury Erskine Programme for supporting D.-S. Byun during his time in New Zealand; to John Thyne for supplying Figure 2 map layers, and to Dr Derek Goring for interesting discussions regarding tidal data sources.

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



**Table 1. Tidal constituent pairs associated with different monthly tidal envelope contributors and their intervals, including their**
**cycle types and controlling factors**

| Constituent pairs | Interval (days) | Cycle type | Control |
|---|---|---|---|
| $M_2$, $S_2$ | 14.7653 | spring-neap | Moon phase, i.e. the axial alignment of Moon and Sun relative to Earth during the synodic month. |
| $M_2$, $N_2$ | 27.5546 | perigean-apogean | Relative distance between the Moon and Earth throughout the Moon's orbit over the anomalistic month. |
| $K_1$, $O_1$ | 13.6608 | tropic-equatorial | Changes in the Moon's declination during the sidereal month. |

*Note*. **With monthly tidal envelope characterization, the $N_2$ is considered in addition to the constituents included in daily tidal form**
**classification (e.g. Defant 1958).**



**Ocean Science** Discussions · Open Access · EGU

**Table 2.** Comparison of tidal constituent amplitudes, amplitude ratios (including daily tidal form factor, $F$, and monthly tidal envelope factor, $F_M^s$) and ranges between the four distinct types of monthly tidal envelope found in the 23 case study semi-diurnal tide regimes of Aotearoa New Zealand, and compared to Equilibrium Theory amplitude ratios

| Monthly tidal envelope description | $F\left(=\frac{K_1+O_1}{M_2+S_2}\right)$ | Amplitudes (cm) of: | | | | | Amplitude ratios of: | | | | | | | | Example sites |
|---|---|---|---|---|---|---|---|---|---|---|---|---|---|---|---|
| | | $M_2$ | $S_2$ | $N_2$ | $K_1$ | $O_1$ | $\frac{S_2}{M_2}$ | $\frac{N_2}{M_2}$ | $\frac{S_2}{N_2}$ | $\frac{S_2+N_2}{M_2}$ | $F_M^s\left(=\frac{M_2+N_2}{M_2+S_2}\right)$ | $\frac{K_1}{M_2}$ | $\frac{O_1}{M_2}$ | |
| n/a | 0.68 mixed diurnal | - | - | - | - | - | 0.47 | 0.19 | 2.44 | 0.66 | - | 0.584 | 0.415 | Equilibrium Theory* |
| Spring-neap type | 0.05 semi-diurnal | 55 | 26 | 9 | 2 | 2 | 0.47 | 0.16 | 2.89 | 0.64 | 0.79 | 0.04 | 0.04 | Kapiti |
| Intermediate, spring-neap dominant | 0.04 to 0.07 semi-diurnal | 78 to 113 | 19 to 29 | 17 to 23 | 2 to 6 | 1 to 4 | 0.24 to 0.27 | 0.18 to 0.22 | 1.12 to 1.45 | 0.45 to 0.46 | 0.93 to 0.98 | 0.02 to 0.06 | 0.01 to 0.05 | Manukau, Westport, Charleston, Puysegur Point North Cape, Boat Cove and Fishing Rock (Raoul Island), Dog Island, |
| Intermediate, perigean-apogean dominant | 0.06 to 0.14 semi-diurnal | 50 to 112 | 4 to 18 | 10 to 22 | 2 to 8 | 1 to 4 | 0.06 to 0.2 | 0.2 to 0.23 | 0.29 to 0.94 | 0.28 to 0.43 | 1.01 to 1.15 | 0.02 to 0.10 | 0.01 to 0.06 | Auckland, Lottin Point, Tauranga, Korotiti Bay, Moturiki, Green Island, Port Chalmers, Sumner, Gisborne, Napier |
| Perigean-apogean type | 0.08 to 0.12 semi-diurnal | 48 to 65 | 2 to 3 | 10 to 14 | 2 to 4 | 2 to 4 | 0.04 to 0.05 | 0.21 to 0.22 | 0.18 to 0.21 | 0.25 to 0.27 | 1.16 to 1.18 | 0.04 to 0.06 | 0.04 to 0.06 | Kaikoura, Owenga, Castlepoint, Wellington |

*Note.* Data sourced from this research, and from Defant (1958)* (for tidal station details, including tidal phase-lags, see Appendix 1 Table A1).






**Table 3. Monthly tidal envelope factor ($F_M^S$) for classifying different monthly tidal types in Aotearoa New Zealand's semi-diurnal**
**tidal regime**

| $F_M^S$ | Type | Name | Description of monthly tidal envelope |
|---|---|---|---|
| $< 0.795$ | 1 | Spring-neap type | Two similar magnitude spring-neap cycles. |
| $> 0.795 \ and < 1.0$ | 2 | Intermediate, spring-neap dominated | Two unequal spring-neap cycles. |
| $> 1.0 \ and < 1.15$ | 3 | Intermediate,          perigean-apogean dominated | A strong perigean-apogean cycle plus two weaker spring-neap cycles. |
| $> 1.15$ | 4 | Pperigean-apogean type | A distinct perigean-apogean cycle. |






**Table 4. Typical (representative) input and boundary values for classifying monthly tidal envelope types around the Aotearoa New**
**Zealand coast**

| $F_M^S$ boundary case | Eq. (1) | | $F_M^S$ | Notes |
|---|---|---|---|---|
| | Given value | $x$ or $y$ | | |
| Type 1 vs. Type 2 | $\dfrac{a_{S_2}}{a_{M_2}} = 0.47$ | $x = 0.36$ | 0.8 | When MSR≈MTR with $F$=0.25, then the value of $F_M^S$; $a_{S_2} = 2.78\, a_{N_2}$ |
| Type 2 vs. Type 3 | $\dfrac{a_{N_2}}{a_{M_2}} = 0.21$ | $y = 1$ | 1 | $a_{S_2} = a_{N_2}$ |
| Type 3 vs. Type 4 | $\dfrac{a_{S_2}}{a_{M_2}} = 0.05$ | $x = 4.024$ | 1.15 | $a_{S_2} = 0.249\, a_{N_2}$ |

*Note.* $x = \dfrac{a_{N_2}}{a_{S_2}}$ and $y = \dfrac{a_{S_2}}{a_{N_2}}$. **MSR and MTR denote the ratio between the maximum spring tide range to the subsequent spring tide**
**range and the ratio between the 'annual' maximum tidal range to the subsequent tidal range, respectively.**



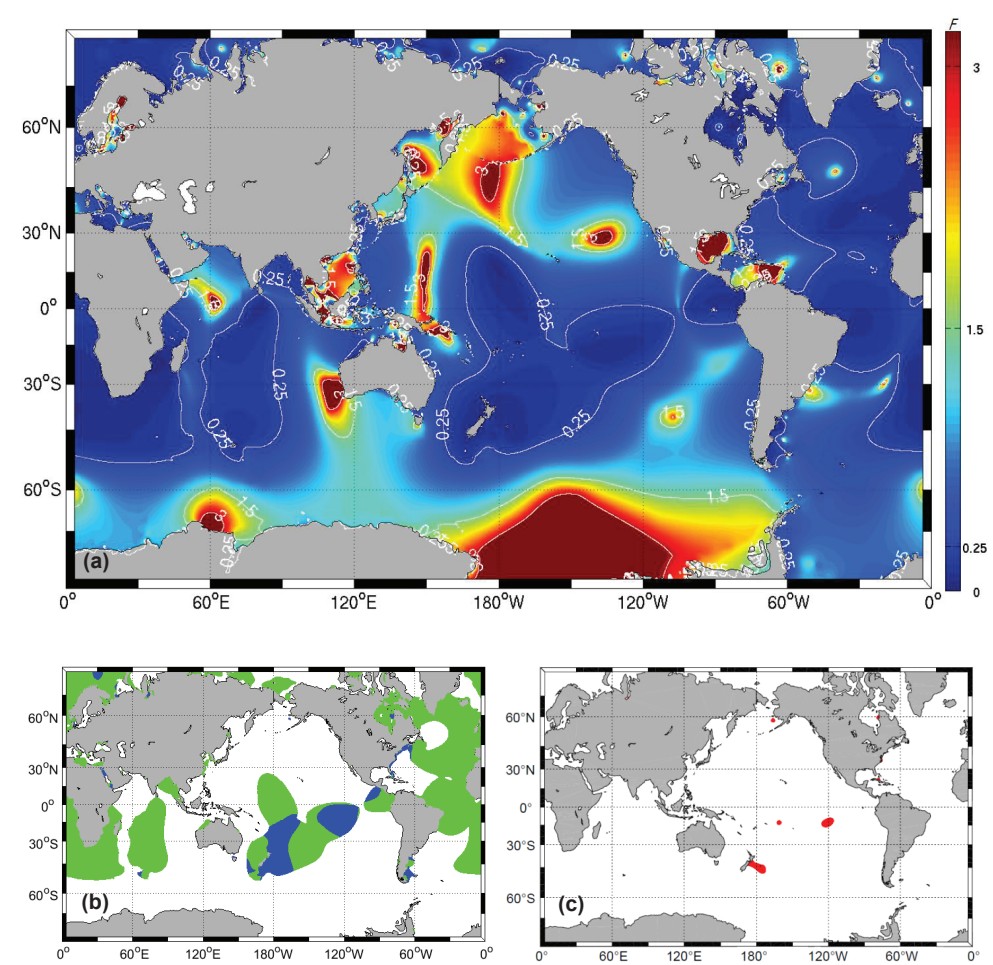

**Figure 1. (a) Global distribution of daily form factor (*F*) values, indicating daily tidal regime types (*F*<0.25: semi-diurnal; *F*>0.25 to *F*<1.5 mixed-mainly semi-diurnal; *F*>1.5 to *F*<3: mixed-mainly diurnal; and *F*>3: diurnal, according to the classification of van der Stok 1897, and Courtier 1938); (b) the world's semi-diurnal tidal areas (*F*<0.25) divided into those where spring-neap (green) versus perigean-apogean (blue) signals are the main influence on the monthly tidal envelope; and (c) semi-diurnal tidal regimes (in red) where the $S_2/M_2$ constituent amplitude ratio is <0.04 and thus spring-neap tidal signals are very weak so that perigean-apogean signals are prominent, as derived from FES2014 tidal harmonic constants.**

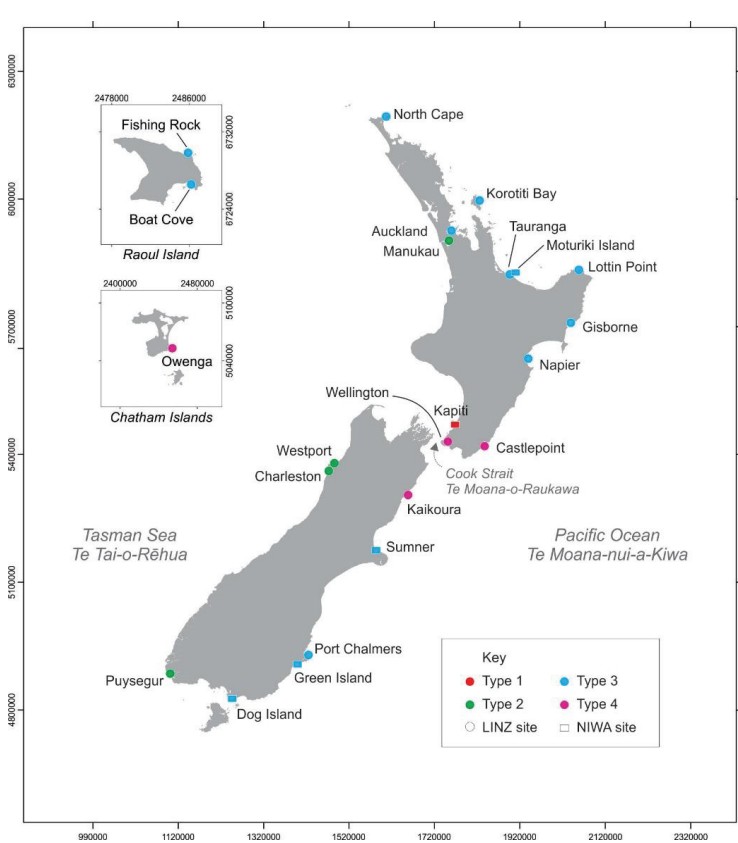

**Figure 2. Location of 23 Aotearoa New Zealand sea level observation stations investigated in this research: circles indicate LINZ**
**sites, rectangles indicate NIWA sites; each site is colored according to monthly tidal envelope type. Offshore islands are not shown**
**to scale (Raoul & Chatham Islands). The coordinate system is NZGD2000, Transverse Mercator.**







Figure 3. Amplitude contours for the (a) $M_2$, (b) $S_2$, (c) $N_2$, (d) $K_1$, and (e) $O_1$ tides around ANZ, and (f) the resultant horizontal distribution of $F$, daily tidal form factor values, as derived and calculated from the FES2014 tide model database at a scale of $1°/16 \times 1°/16$. Note that the amplitude color scales vary between plots a and e.

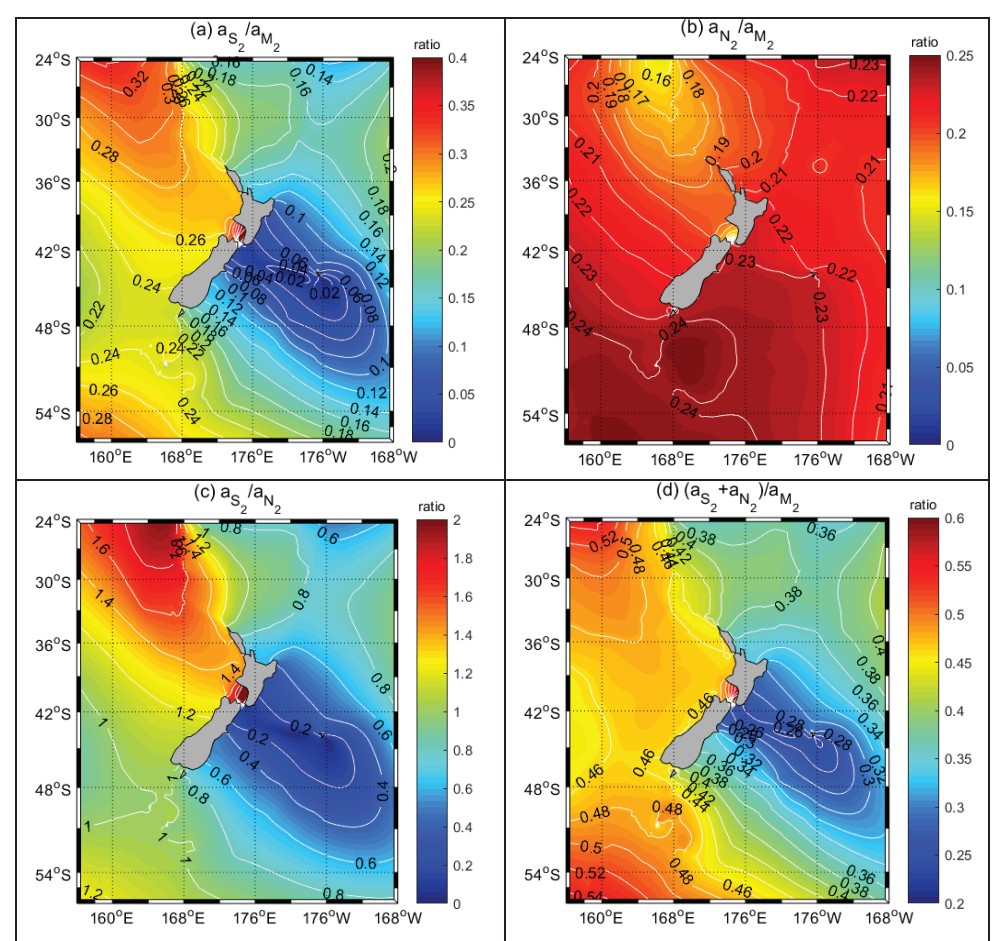

**Figure 4. Horizontal distributions of tidal constituent amplitude ratios around ANZ for: (a)** $\frac{a_{S_2}}{a_{M_2}}$**; (b)** $\frac{a_{N_2}}{a_{M_2}}$**; (c)** $\frac{a_{S_2}}{a_{N_2}}$ **and (d)** $\frac{a_{S_2}+a_{N_2}}{a_{M_2}}$**; as calculated using the FES2014 tide model database at a scale of 1°/16×1°/16. Note that the amplitude color scales vary between plots a and d.**


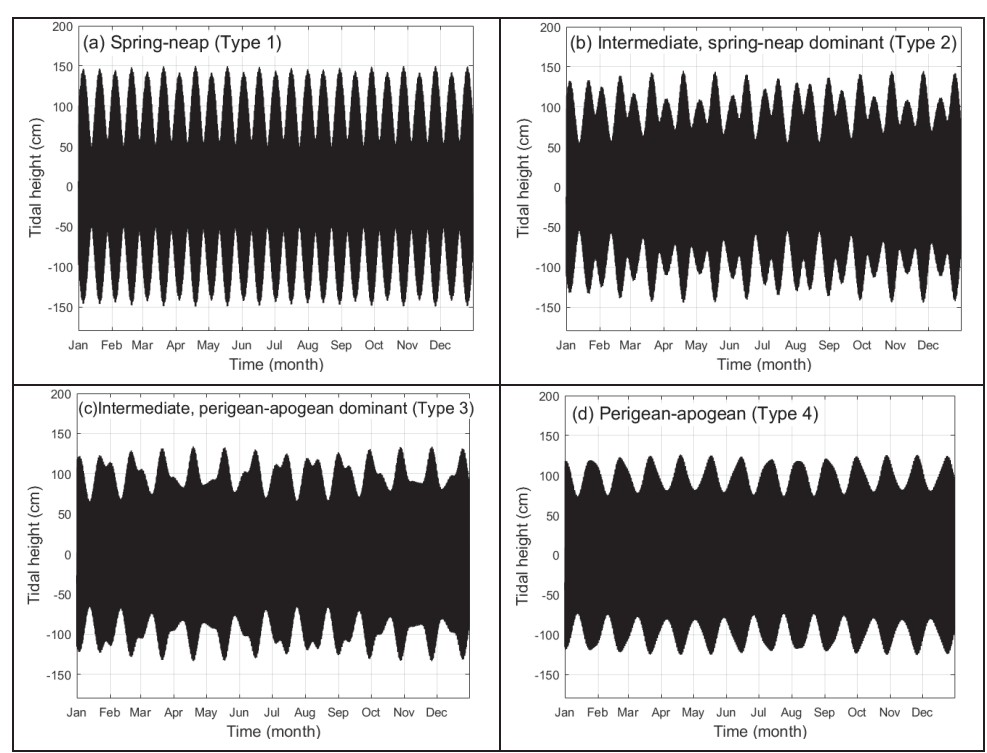

**Figure 5. Idealized examples of four different monthly tidal envelopes over one year, calculated using the amplitude value $a_{\mathrm{M_2}} = 100\ cm$ and the amplitude ratio values of: (a) $\frac{a_{\mathrm{S_2}}}{a_{\mathrm{M_2}}} = 0.46$, $\frac{a_{\mathrm{S_2}}}{a_{\mathrm{N_2}}} = 11.5$, $\frac{a_{\mathrm{N_2}}}{a_{\mathrm{M_2}}} = 0.04$; (b) $\frac{a_{\mathrm{S_2}}}{a_{\mathrm{M_2}}} = 0.27$, $\frac{a_{\mathrm{S_2}}}{a_{\mathrm{N_2}}} = 1.5$, $\frac{a_{\mathrm{N_2}}}{a_{\mathrm{M_2}}} = 0.18$; (c) $\frac{a_{\mathrm{S_2}}}{a_{\mathrm{M_2}}} = 0.12$, $\frac{a_{\mathrm{S_2}}}{a_{\mathrm{N_2}}} = 0.5455$, $\frac{a_{\mathrm{N_2}}}{a_{\mathrm{M_2}}} = 0.22$; and (d) $\frac{a_{\mathrm{S_2}}}{a_{\mathrm{M_2}}} = 0.04$, $\frac{a_{\mathrm{S_2}}}{a_{\mathrm{N_2}}} = 0.1818$, $\frac{a_{\mathrm{N_2}}}{a_{\mathrm{M_2}}} = 0.22$. Note that the $F_M^S$ values of these plots are: (a) 0.71; (b) 0.93; (c) 1.09; and (d) 1.17.**





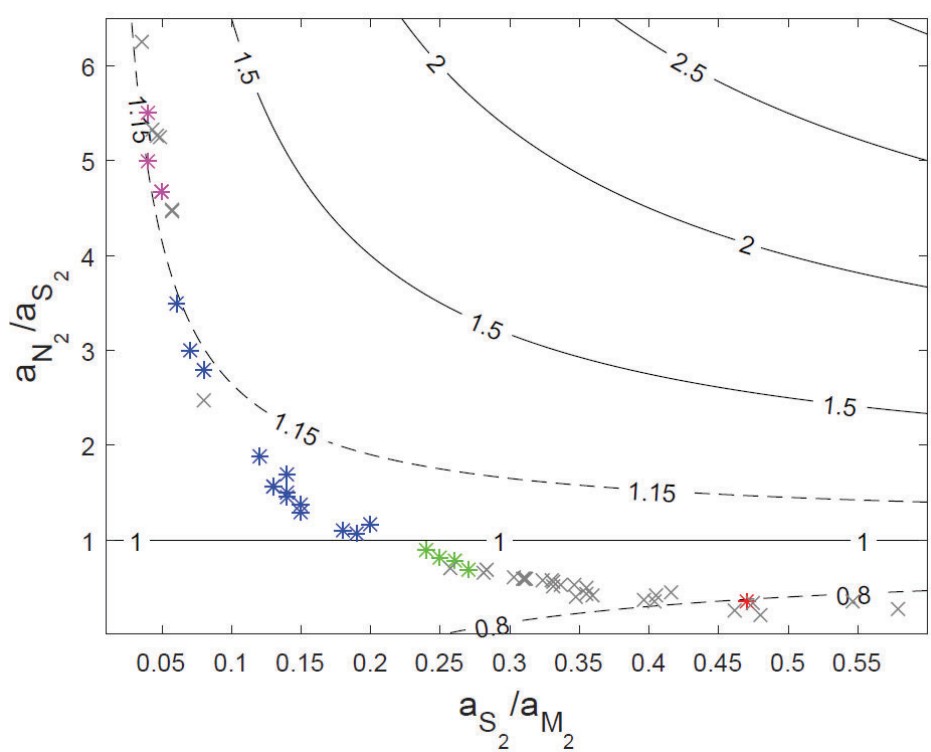


**Figure 6. Plot of the relationship between the $\frac{a_{N_2}}{a_{S_2}}$ and $\frac{a_{S_2}}{a_{M_2}}$ ratios (y and x axes respectively) versus $F_M^S$ values (shown as plot contours),**
**with data points corresponding to Aotearoa New Zealand waters Type 1 sites (red star); Type 2 sites (green stars); Type 3 sites (blue**
**stars); and Type 4 sites (pink stars), all from Table 2; and tidal data representative of the greater Cook Strait area (grey crosses)**
**from Walters et al. (2010, Tables 1 and 3).**






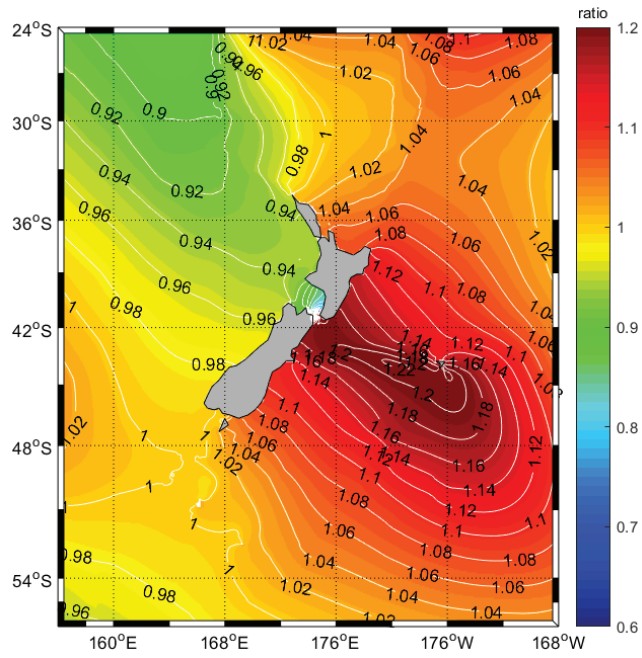


Figure 7. Distribution of monthly tidal envelope factor ($F_M^S$) values in the waters around ANZ, calculated using FES2014 data. See
Table 3 and Figure 5 for corresponding monthly tidal envelope factor classes and envelope patterns.