# Peer review of "A monthly tidal envelope classification for semidiurnal regimes in terms of the relative proportions of the S2, N2, and M2 constituents"

_Ocean Science, 2019_

## Referee Comment (RC1) · Philip Woodworth (Referee) · 22 Dec 2019

December 2019

Comments on 'A monthly tidal envelope classification approach for semi-diurnal regimes with variability in S2 and N2 tidal amplitude ratios' by Byun and Hart (OSD)

New Zealand has a predominantly semidiurnal tide all around its coast. However, there is a major difference between the west and east coasts. On the west coast, the semidiurnal tide varies over a fortnight (spring-neap) as it does in most parts of the world. However, the tides of the east coast of NZ are unusual in having a larger

contribution from the N2 constituent than S2, with the result that the tidal range varies over a month rather than a fortnight. This aspect has complicated an understanding of the local tides since the first measurements made by James Cook (Woodworth and Rowe, Hist. Geo and Space Sciences, 2018). Consequently, the authors suggest a new type of 'form factor' to indicate the extent to which the semidiurnal tide is 'normal' (i.e. spring-neap variation over a fortnight) or 'unusual' (i.e. varies over a month).

The idea behind the work is not a very profound one but I can see that the new form factor could be useful to coastal studies in NZ by providing a first-order description of tidal behaviour. Therefore, I have nothing against the paper as such, although I am surprised that NZ tidal scientists have not come up with similar classifications before. The paper is a short one and could be much shorter if the text was not so repetitive. I make some comments below which I hope are of use for a second draft. Many are trivial ones to do with the text and can be easily attended to, while a couple are more important.

Abstract - if I had written this abstract I would have used the useful words on lines 206-214 of the Discussion. For example, I can see that the new form factor could inform about shoreline ecology as ecology depends on the tidal climatology. However, I cannot see that it is much use in discussion of inundation hazards and climate change; for that one would be interested primarily in the character of sea level extremes and not just on simple descriptions of the tide.

- remove 'database'. 'theoretical experiments' –> 'theoretical arguments' maybe.

- the symbol Fsm is a clunky one and even impossible to write on an ascii keyboard. What is it supposed to mean? A form factor showing S2's influence on M2? But what about N2 i.e. Fnm? I would have invented a simpler symbol such as F-prime or maybe E for envelope?

- I don't see that the first two references are really relevant to this sentence. Cartwright is a history of tidal science. D'Onofrio discusses Buenos Aires only and not spatial variation. The Nicholls reference is ok.

- 'and gravimetry'. What does that refer to? Space gravimetry by missions such as GRACE? I would drop that. Then again the references are apparently random - Egbert et al. describes one particular model, while Stammer et al. describes many including Egbert. So why is Egbert here and not all the others?

- I would have the equation here i.e. F=(K1+O1)/(M2+S2) and not just words, like your equation (1) below which would become (2)

26-27 - if you have four you can't add a fourth?

- aren't they the same form factor (singular)?

I am not familiar with the van der Stok and Courtier references which are very old and I don't think many other readers will be either. How did you come across them? If in a more recent history of tides or a text book on tides then please add that.

34-36 this is a garbled sentence. Could you please reword?

45-47 This isn't right. You say yourself that NZ tides are unusual so the reviews of Andersen etc. cannot be blamed for focusing on the main constituents relevant to global studies. However, that does not mean those authors were disinterested in other constitents. In fact one main aim of such studies was to determine how well the total tide could be determined which necessitates accuracy in N2 etc.

- as mentioned above I can't see form factors (of whatever kind) being directly relevent to coastal flooding hazards work, but if I am wrong please give references.

- why don't you just have a simple map here for the reader to refer to i.e. Figure 2, and not wording such as 'latitudinal gradient' - you mean range of latitude. Having the Type information in the figure is ok but you have to return to that later (see below about that)

- what are 'absolute tides'?

68-69 - this business of a pair of amphidromes to the NW and SE is not easy for the reader to appreciate from your wording alone, and the amphidromes are in fact a long way NW and SE and off the maps of Figure 3. So you have to point the reader to where he can see a map of M2 in the SW Pacific - ideally a map from FES2014 as you have focused on that. Or see Fig 5.1 of Pugh and Woodworth (2014) which was provided by Richard Ray - i.e. a wider area than you have used for Fig.3.

Anyway I don't think it is right to say S2 has a single wavefront and amphidrome in the SE. Take a look at Figure 4 of Walters et al. (2001) and you will see a pair of them close together in the SE.

And I would drop mention of the Coriolis effect and simply say that they rotate anticlockwise.

- 'years' is misleading as it suggests you have used many years per station whereas Table A1 shows you used only one year for each. Have the amplitudes and phase lags in Table A1 been adjusted for nodal variations according to equilibrium relationships? Or are they the observed amplitudes for the years shown? See below for other comments on this table.

I would have prioritised the FES2014 model over the tide gauge data as the main aspects of what you are trying to show are best done with the model. Then at the end of the paper you can show your findings from FES2014 are consistent with those from the tide gauge positions.

- you mean 'in comparison with values obtained from the tidal potential or Equilibrium Tide'

- 'amplitude data' –> 'amplitudes'. 'was sourced' –> 'was obtained'

- days' length or days in length

- tides are the strongest

85, 242 and 268 - Carrere has an accent over the first e

- dataset –> model. experimental plots –> studies (maybe)

- siderial –> diurnal.

92-93 - mapped spatial variability

94-95 - .. those from the Equilibrium Tide (Defant, 1958).

It was not Defant's theory. Anyway you might better refer to Cartwight and Tayler (1971) for example.

- 'data results' –> results

- reinforces –> shows

- .. amplitude (Figure 3c).

- in the text and tables and figures it would be much simpler if you dropped the 'a' and have M2 for example to refer to its amplitude. All the a's make things messy. You would have to say you were doing that of course.

- drop relatively

The two bullets below. Could you mention them as determining Type 1 and Type 4.

- surely that is not referring to Figure 1, you can't see Cook Strait in that at all

- 75 to 90% of what? What are the adjacent coasts?

- 'anomalistic timeframes' –> 'a month' and drop the 27.5546 four decimals –> 27.6 will do

Chatham Rise and Castle Point are not in Figure 2.

sentence 'By examining'. I would drop this sentence. You repeat yourself a lot.

- I would say spring-neap and then perigean-apogean as that is the order elsewhere

Two bullets. Can you mention them as Type 2 and 3

- amplitudes being only

- Sumner not in Figure 2

Equations 1 - drop the a's (see above). Also drop the 'more stable' words. I guess you mean similar locally? But the same situation would apply if the constituents varied a lot spatially. These are just simple algebraic relationships at a particular position - they have nothing to do with spatial scale or 'stability'.

Table 4 should be 3?

- it is strange to read of M2, which is the largest, moderating something smaller. I think this paragraph needs rewording. Also I don't understand (i) and (ii). What are the 'annual' and 'subsequent' things? I guess the R in MTR is ratio? Not clear. But note that MTR stands for Mean Tidal Range in usual tidal studies.

Anyway I found these experiments at lines 180-204 somewhat unconvincing, although I do understand why you felt the need to inject some rigour into the choice of boundary values between Types. But the experiments do not cover the whole space of possibilities for amplitude and phase lag of all constituents concerned. The main thing to me is Figure 6 which shows nicely how Fsm varies with x and y. Why don't you then just define the boundaries between Type 1 etc. in an ad hoc way, similar to the way as F is divided in an ad hoc way for 'semidiurnal' etc. After all, in the end all these form factors are just handy coarse descriptive subdivisions for the tide. Anyway lines 180-204 need rewriting - see my comment at line 173 also. It is just not clear what you are doing.

- see below. mention the other red blob.

- if you agree then drop 'theoretical experiments' here.

- these three do not all operate at 'synodic anomalistic timescales'. Why not just '..

three key constituents (M2, S2 and N2).'

At this point it occurred to me that a similar exercise could be conducted for areas of predominantly diurnal (but a bit mixed) tides. Could you speculate in this Discussion which parts of the world could benefit that way?

- this isn't necessarily true. Figure 1c shows where S2 is small compared to M2. It doesn't necessarily follow that perigean influences dominate.

- what is 'low-frequency coastal flooding'?

Table A1. Line 1 - you don't show tidal ranges, this will be confusing for most people. What you show on the last line are ranges of amplitudes and phase lags in your data set. Also the 'ranges' shown are crazy for some as shown e.g. see 6-360 for K1. But 360 degrees is the same as 0 degrees! Line 2 - values. Also the header should mention you show Types. Say if the phase lags shown are in Greenwich Mean Time or local time? if Greenwich then they are usually denoted by G.

Figure A1. I don't understand the 'under conditions summarised in Table A1'. Surely all one needs to know is which stations were used for these 3 examples.

- doesn't matter much but Figure 2 looks like a simple coastline map to me that one could make with GMT or Matlab, so where do the fancy 'map layers' come in? And with an undesirable national coordinate system to boot instead of lat/lon?

Table 1. Line 1. The word 'interval' in tides refers to the times of high tide since passage of the moon. What you are showing here are not intervals but the periods of beating of the shown pairs of constituents. And personally I would abandon columns 3 and 4 - you are not writing a text book here - certainly drop column 3 (and in M2/S2 - drop 'axial'. M2/N2 - drop 'relative'. line 3 'during the siderial month' –> during a month). And I would drop the Note which doesn't add anything.

Table 2. I would have a column 1 showing Type. And I would move Example Sites to be a column 2. First line of that: Equilibrium Theory (no footnote and no Note - you have already mentioned Defant in the text).

Table 3 - I guess this does no harm but it just repeats what has been given in the text. I would drop it. Pp..

Table 4 - I don't understand this table. It is tied up with mention of the experiments, see comments above. I would drop this table as well.

Figure 1 - just a suggestion but perhaps all panels could be made the same size. You have (a) large but that is for the normal F which is not the subject of this paper and can be found in many text books.

Also for this, and also for the other colour maps in Figs 3,4 etc. could you have an arrow on the max colour as you have points on the maps with values which are in overflow.

As for Figure 1 (c), you should mention somewhere in the text where the other red blob is. Near Tahiti?

line 4 of caption '.... monthly tidal envelope using criteria described in section 3.' Then for (c) see my comment for line 230.

Figure 2 - please use conventional lat/lon and not a national coordinate system no-one else will understand. As mentioned above there are places in the text (e.g. Stewart Is.) not shown. '&' –> 'and'.

When Figure 2 is first mentioned in the text there is no mention of the Type 1, 2 etc. So you have to returrn to this figure after you discuss Figures 6 and 7 and mention the Types in Fig.2, and then please also use the same colours for the Types here as in Fig 6.

Fig 3 - arrows needed on colour scales e.g. for the overflow top-left of 3(d). The contour annotation bottom right of 3(f) is messy, please thin out the annotations. Also drop 'Unit' in 'Unit mm'.

line 1 of caption - 'Amplitudes for'. Drop 'horizontal'. Line 2 - drop 'derived and'. drop

'database'. 'at a scale of' –> 'on a grid of'

Fig 4 - as mentioned I would drop the a's in the headers and captions. Arrows on colour scales.

line 1 of caption - drop 'horizontal'. Line 2 - drop 'database'. 'at a scale of' –> 'on a grid of'

Fig 5 - drop a's

Fig 6 - this is actually a useful plot. Use another colour instead of pink which is too much like red. drop a's. Use same colours for the Types as in Fig 2.

Add dotted or dashed lines also for the Fsm boundary values chosen to define Types 1-4.

Also what would be useful also would be to have values from FES2014 for the whole NZ coastline - that might be a fiddly computing exercise but is obviously possible.

Fig 7 - overflow arrow. could roughly the same colours be used as for Fig 6 as far as possible? That has red-green-blue-pink for types 1-4 whereas this has green-yellow-red more or less (the blue is not used).

line 2 - .. see Figure 5 for definitions and examples of ..

---

## Short Comment (SC1) · 20 Jan 2020

Place names: There is some inconsistency in the place names used in the paper. For example, Cook Strait is referred to as 'Cook Strait' (line 73 and others) and 'Te Moana-o-Raukawa Cook Strait' (line 70 and Figure 2). Lines 60 and 61 give both alternative names for the North Island and South Island. The English and Maāri names are alternatives; it is not necessary to use both - choose one form and use it consistently. The official name for Stewart Island is 'Stewart Island / Rakiura', not 'Rakiura or Stewart Island' as shown in line 61. It is recommended that place names used are as shown in the NZ Place Names Gazetteer. Cook Strait is just 'Cook Strait' (not an official names

but a recorded one), 'Castle Point' (line 117) is 'Castlepoint' (official). 'Aotearoa New Zealand' has been used for the name of the country (and abbreviated to ANZ) but until an Act of Parliament is passed the country is 'New Zealand'.

Line 82: It would be helpful to point out that the results of the analysis of the records from the additional 33 locations are presented in Figure 6, and a sentence or two summarising those results would be appropriate.

Line 83: What does 'reach the strongest' mean?

Line 93 and 94: The text here states that Figures 3 and 4 map the constituent amplitudes and ratios listed in Table 1, but surely the figures are derived from the FES2014 model as stated in the captions for Figures 3 and 4.

Line 117: Castlepoint is on the east coast of the North Island.

Line 125: What does 'combined variability' mean? The rest of this sentence is difficult to follow - a diagram might help?

Figure 2: Castlepoint is shown out of its true position.

---

## Author Comment (AC1) · 20 Jan 2020

We are very grateful for this review as it has been useful in helping to improve our paper. Below we have copied each individual reviewer comment, and written below it a response. Almost all suggested changes have been adopted wholesale, but discussion and a couple of questions remain below regarding the link made to flooding hazard.

Individual reviewer comments and our responses

Abstract - if I had written this abstract I would have used the useful words on lines 206-214 of the Discussion. For example, I can see that the new form factor could inform about shoreline ecology as ecology depends on the tidal climatology. However, I cannot see that it is much use in discussion of inundation hazards and climate change; for that one would be interested primarily in the character of sea level extremes and not just on simple descriptions of the tide. Response: Thank you for this suggestion regards the discussion text – we have used some of this text to replace the original opening sentence of the abstract. Regards the inundation hazards and climate change link drawn here regarding perigean-spring tides, this comment relates our experiences in Christchurch (e.g. Allen et al., 2014; Hart et al., 2018). This city (marked by the Sumner gauge site in Fig. 2) experienced up to 1 m relative sea level rise in coastal and river proximal suburbs due to subsidence during the Canterbury Earthquake Sequence (CES, 2010-2011). This instantaneous sea level change was equivalent in magnitude to that which had been predicted (in absolute as opposed to relative terms) for the next 50 to 100 years due to anthropogenic climate change and accelerated sea level rise. We thus use Christchurch as a 'laboratory' to consider what 1 m of sea level rise might look and feel like in a delta city (of which there are many similar settings in seismically active areas worldwide), albeit with process-response timescales being rather different to those under climate change scenarios. One of the greatest effects has been enhanced flooding issues, much more so than other coastline hazards such as erosion. Since the city relies on river and estuary drainage conduits, in particular, when pronounced perigean-spring tides occur or combination with sustained rainfall events, inland riverside and low-lying coastal suburb flooding is widespread, deep and persistent. It would seem that around half of the city had little freeboard, and that buffer has been significantly reduced with the CES such that even monthly high tides pose issues for the lowest lying areas nowadays. The backwater effects of high tides combine with atmospheric low pressure and sustained precipitation events to extend the reach of flooding. We suspect that the latter will continue to be enhanced as the baseline mean sea level rises with climate change, meaning less ability to cope with perigean-spring tides. Since the CES, high tide alerts have become of wider public interest since they are now commonly associated with flooding. Understanding the frequency of such tidal alert days has been of use to those at the coal face of flooding, in terms of emergency responses as well as in making decisions about whether to stay or retreat from subsidence affected areas. We therefore see monthly tidal height patterns as intricately linked to questions of initial sea level rise effects in our city.

- remove 'database'. 'theoretical experiments' –> 'theoretical arguments' maybe. Response: Both changes have been made.

- the symbol Fsm is a clunky one and even impossible to write on an ascii keyboard. What is it supposed to mean? A form factor showing S2's influence on M2? But what about N2 i.e. Fnm? I would have invented a simpler symbol such as F-prime or maybe E for envelope? Response: As suggested, we have changed F_MˆS throughout the paper to the much simpler notation of E.

- I don't see that the first two references are really relevant to this sentence. Cartwright is a history of tidal science. D'Onofrio discusses Buenos Aires only and not spatial variation. The Nicholls reference is ok. Response: The first two references have been removed from the sentence: "Successful human-coast interactions in the world's low-lying areas are predicated upon understanding the temporal and spatial variability of sea levels". Please note that we meant the phrase 'temporal and spatial variability in sea levels' to encompass a wide range of processes including cyclical tidal height variations, and were not meaning mean sea level variations alone, a topic best highlighted using the third reference. We have added the reference Woodworth et al. (2019) to emphasise this wider meaning. This view of sea level variability was why we had Cartwright's book on tides here, because it covers the spatial aspect of sea level variability. Regards D'Onofrio et al. (1999), although this paper case studies just one place (Buenos Aries), it is a nice example of a low-lying coastal city study that combines analyses of storm surge and tidal height probability distributions, with consideration given to the importance of a robust temporal analysis of each element to predict the frequencies of future sea level extremes. This paper finishes by pointing out that the temporal dimension of these results will adjust in future with sea level rise.
Such a careful multi-hazard approach is lacking in many studies, hence our desire to highlight this work.

- 'and gravimetry'. What does that refer to? Space gravimetry by missions such as GRACE? I would drop that. Then again the references are apparently random – Egbert et al. describes one particular model, while Stammer et al. describes many including Egbert. So why is Egbert here and not all the others? Response: We have removed "and gravimetry" and the Egbert et al. (1994) reference. This reference was in there as a succinct example of a research paper that focusses on removing tidal signals from sea levels, in their case from TOPEX POSEIDON altimeter data. We used this reference specifically as it was an example of tidal signal removal based on just the four main constituents that comprise F. Stammer et al. (2014, p243) stated the point, which we repeated in shorter form, that "An especially important application for accurate tide models is providing tide "corrections" to various measurements so that smaller nontidal signals may be studied. For example, barotropic tide models are used regularly to remove tidal variability from space geodetic observations; this is a critical necessity for successful satellite altimetry [e.g., Fu and Cazenave, 2001] and satellite gravimetry [Seeber, 2003; Visser et al., 2010], and in both cases improved tidal corrections lead to a reduction of aliased tidal "noise" in nontidal signals of interest".

- I would have the equation here i.e. F=(K1+O1)/(M2+S2) and not just words, like your equation (1) below which would become (2) Response: We have added this equation explicitly, as suggested, and renumbered the other equation.

26-27 - if you have four you can't add a fourth? Response: This has been clarified in the text. Originally van der Stok (1897) divided tidal regimes into three types using the F equation, while Courtier (1938) added a fourth (daily) tidal regime type.

- aren't they the same form factor (singular)? Response: This has been corrected in the text.

I am not familiar with the van der Stok and Courtier references which are very old and

I don't think many other readers will be either. How did you come across them? If in a more recent history of tides or a text book on tides then please add that. Response: Van der Stok (1897) was available to us via interlibrary loan. We borrowed an original 1897 large format book from California through the UC library – see available copies here: https://www.worldcat.org/title/wind-and-weather-currents-tides-and-tidal-streams-in-the-east-indian-archipelago/oclc/488220907. It is an interesting piece of work as it clearly outlines the F equation and three of the four tidal regime types in common usage today, in a work dating back over 120 years. We feel that it is best to leave this reference in our paper discussion of the origins and history of use of F, not least to give credit to this early author. We found out about van der Stok's work from its citation in Courtier (1938), which is available online, in a PDF English translation, from: https://journals.lib.unb.ca/index.php/ihr/article/download/27428/1882520184. We have added this web link in our reference list entry for Courtier (1938) to make it more accessible to readers. Both of these references were located via a Google search (in contrast our university library multi-search returned no useful results).

34-36 this is a garbled sentence. Could you please reword? Response: We have reworded this sentence into two shorter, clearer sentences as follows: The daily tidal form factor identifies the typical number (1 or 2) and form (equal or unequal tidal ranges) of tidal cycles within a lunar day (i.e. 24 hours and 48 minutes) at a particular site. In contrast, the term 'tidal envelope' describes a smooth curve outlining the extremes (maxima and minima) of the daily tidal cycles occurring at a particular site through a specified period of time.

45-47 This isn't right. You say yourself that NZ tides are unusual so the reviews of Andersen etc. cannot be blamed for focusing on the main constituents relevant to global studies. However, that does not mean those authors were disinterested in other constitents. In fact one main aim of such studies was to determine how well the total tide could be determined which necessitates accuracy in N2 etc. Response: We have deleted these lines.

- as mentioned above I can't see form factors (of whatever kind) being directly relevent to coastal flooding hazards work, but if I am wrong please give references. Response: You are absolutely right to question the unusual link drawn in our paper between form factors and coastal flooding hazards work, and we cannot provide you with a published tidal reference that neatly encapsulates this idea directly and fully. However, I would like here to offer some explanation for why we think this link exists for some places (delta cities) and is relevant, then to seek your expert advice on whether it would be best to delete the link from our current paper, or to try to express the idea more clearly for our readers. Again using Christchurch city as an example - this city is situated towards the centre of Aotearoa New Zealand's east coast region of strongly perigean-apogean influenced tides. The city is constructed (like Tokyo, Jakarta, Charleston NC and many other delta cities) on a low-lying, formerly swampy, coastal progradation and river delta plain in a seismically active area. This physical setting, combined with imprudent development, has influenced the flood hazard: major flooding occurs when periods of sustained heavy (as opposed to high intensity) rainfall produce river and overland flows which fail to drain efficiently through the city's distributed, gravity-based and sea level connected drainage network (Hart et al., 2018). One of the key factors that determines whether or not a sustained rainfall event will result in widespread and severe flooding, or not, is the tides. Flooding is more likely during perigean tides, since these times feature periods of more than a week with particularly high tidal ranges. As illustrated in the Fig. R1 (top), unlike in spring-neap dominated areas, periods of high tidal ranges in Christchurch can last for well over a week with short duration periods of smaller range tides between, when flooding is less likely. This means that high tide 'red alert' days (Fig. R1 (below)) can last for more than a week, and there is an increased chance that these might coincide with sustained rainfall events, than in more spring-neap dominated regions which feature the distinct and regular punctuations of the lower range neap tides. This is a subtle but genuine reason why we believe it is important for 'delta cities' like Christchurch to consider their monthly tidal pattern when considering the multiple factors that influence flooding. An additional aspect of this idea relates to how we quantify future flood risks and return periods under changing climate (not to mention in the multi-hazard context of future seismic activity, e.g. Allen et al., 2014). The tidal height patterns will not be hugely influenced by climate change, so we can already produce accurate frequency histograms and probability distributions for future high tide levels, like that conducted by D'Onofrio et al. (1999) for Buenos Aries. Future rainfall and storm surge statistics are harder to predict under changing climate and need to be combined with the more predictable tidal water level contributions to establish accurate flooding and inundation risk predictions. In the past we in ANZ focussed on flood return periods established using historical water level records, but this is no longer a robust practice since the more predictable tidal water level probabilities need to be combined with the changing atmospheric components to produce altered flood risk estimates for the future. All this is in a relatively newly colonised country where hydrological data records are short. Our point is partly that amongst all this uncertainty, at least the tidal pattern component of these hazards is nicely predictable, so we encourage colleagues to take tidal patterns into account in their flood hazard predictions (something that has been lacking in past analyses). We hope to make the case for the connection between tidal envelope pattern and flood hazards in an upcoming ASCE (2020) monograph paper on flooding and inundation multi-hazards, but realise that this idea is only hinted at in a very truncated manner in our current paper. Please do recommend if we should delete text making this link in our current paper, or if we should leave it in, albeit as a fleeting mention, or some other suggestion.

- why don't you just have a simple map here for the reader to refer to i.e. Figure 2, and not wording such as 'latitudinal gradient' - you mean range of latitude. Having the Type information in the figure is ok but you have to return to that later (see below about that) Response: We have corrected the word 'gradient' to 'range' and added a citation pointing the reader to Fig. 2 at this point in the text. Also, in Results section 3.1 we have added another mention of Fig. 2, highlighting at this stage the observation station colour coding of their identified monthly tidal envelope types.

- what are 'absolute tides'? Response: We have amended the sentence to read
". . .micro through to macro tidal ranges".

68-69 - this business of a pair of amphidromes to the NW and SE is not easy for the
reader to appreciate from your wording alone, and the amphidromes are in fact a long
way NW and SE and off the maps of Figure 3. So you have to point the reader to where
he can see a map of M2 in the SW Pacific - ideally a map from FES2014 as you have
focused on that. Or see Fig 5.1 of Pugh and Woodworth (2014) which was provided by
Richard Ray - i.e. a wider area than you have used for Fig.3. Anyway I don't think it is
right to say S2 has a single wavefront and amphidrome in the SE. Take a look at Figure
of Walters et al. (2001) and you will see a pair of them close together in the SE. And I
would drop mention of the Coriolis effect and simply say that they rotate anticlockwise.
Response: Regards the description of the M2, we have added a citation Pugh and
Woodworth (2014) Fig. 5.1 – thank you for this suggestion. We have amended our
description of the S2 and K1 tide amphidromes in line with Walters et al. (2001; 2010).
Mention of the Coriolis Force has been removed as suggested.

- 'years' is misleading as it suggests you have used many years per station whereas
Table A1 shows you used only one year for each. Have the amplitudes and phase lags
in Table A1 been adjusted for nodal variations according to equilibrium relationships?
Or are they the observed amplitudes for the years shown? See below for other com-
ments on this table. I would have prioritised the FES2014 model over the tide gauge
data as the main aspects of what you are trying to show are best done with the model.
Then at the end of the paper you can show your findings from FES2014 are consistent
with those from the tide gauge positions. 80 - you mean 'in comparison with values
obtained from the tidal potential or Equilibrium Tide' Response: We have added the
adjective "individual" to highlight that an individual year was analysed for each of the
observation stations, with the text now reading "an individual year of good qual-
ity hourly data was selected for analysis per site from amongst the multi-year records.
The 27 individual year sea level records were then harmonically analyzed using T_Tide (Pawlowicz et al., 2002) with the nodal modulation correction option, to examine spatial variation in the main tidal constituents' amplitudes, phase-lags, and amplitude ratios between regions (see Table A1 for raw results) and to compare them with values obtained from the tidal potential or Equilibrium Tide". In our paper we have chosen to initially employ the analysis of observational data, later on using the FES2014 model data to check our findings and extend our spatial coverage. Although the observational data does not have the same uniform coverage of the FES214 data, we prefer to initially use the observational data since it represents real in situ records that are accurate at the coast, while the FES2014 data is helpful for our final classification around the whole country (i.e. Figure 7).

- 'amplitude data' –> 'amplitudes'. 'was sourced' –> 'was obtained' 82 - days' length or days in length 83 - tides are the strongest 85, 242 and 268 - Carrere has an accent over the first e 86 - dataset –> model. experimental plots –> studies (maybe) 88 - siderial –> diurnal. 92-93 - mapped spatial variability Response: All changes made as suggested. The accent has been added everywhere that Carrère's work is cited.

94-95 - .. those from the Equilibrium Tide (Defant, 1958). It was not Defant's theory. Anyway you might better refer to Cartwight and Tayler (1971) for example. Response: We have altered the misplaced reference to Defant (1958) to indicate that this was the source from which we obtained the Equilibrium Tide data, and not the source of Newton's Theory itself, and we have also deleted the reference to Defant (1958) from the table (now numbered Table 1) since it is now clearer in the text.

- 'data results' –> results 98 - reinforces –> shows 101 - .. amplitude (Figure 3c). 102 - in the text and tables and figures it would be much simpler if you dropped the 'a' and have M2 for example to refer to its amplitude. All the a's make things messy. You would have to say you were doing that of course. 103 - drop relatively The two bullets below. Could you mention them as determining Type 1 and Type 4. 109 - surely that is not referring to Figure 1, you can't see Cook Strait in that at all 110 - 75 to 90% of what? What are the adjacent coasts? 114 - 'anomalistic timeframes' –> 'a month' and drop

the 27.5546 four decimals –> 27.6 will do Response: All changes made as suggested, Figure 1 has been corrected to Fig. 2, and for 110, the text has been amended to "compared to on the western coasts both north and south of this central ANZ area".

Chatham Rise and Castle Point are not in Figure 2. Response: The text read: "This type of regime occurs, for example, around the northern Chatham Rise near Kaikoura, and as far north as Castle Point on the east coast of the South Island". We have left this text and the Fig. 2 map unchanged regards these labels as both Castel Point and Kaikoura were on this figure (see north and south of Cook Strait on the central east coast). Other labels that were missing from Fig. 2 as pointed out below (e.g. North/ South/ Stewart Island) have been now added though.

sentence 'By examining'. I would drop this sentence. You repeat yourself a lot. Response: Deleted – thank you, we really appreciated your suggestions for shortening our text and removing repetition.

- I would say spring-neap and then perigean-apogean as that is the order elsewhere Two bullets. Can you mention them as Type 2 and 3 133 - amplitudes being only Response: All changes made as suggested.

- Sumner not in Figure 2 Equations 1 - drop the a's (see above). Also drop the 'more stable' words. I guess you mean similar locally? But the same situation would apply if the constituents varied a lot spatially. These are just simple algebraic relationships at a particular position – they have nothing to do with spatial scale or 'stability'. 161 Table 4 should be 3? Response: For 139, Sumner is in Fig. 2, on the central east coast of the south island (just above the peninsula – this is a gauge site for Christchurch city). For Eqs. 2, 2a and 2b (formerly 1, 1a and 1b), all 'a's have been deleted and it has been specified in the text that these ratios refer to amplitudes. All references to stability have been dropped. Re '161', both Tables 3 and 4 now seemed superfluous after re-arranging Table 1 and with removal of the experiments, so these two tables (3 and 4) have been deleted, as has all mention of them, shortening our paper nicely (thank you for those suggestions).

- it is strange to read of M2, which is the largest, moderating something smaller. I think this paragraph needs rewording. Also I don't understand (i) and (ii). What are the 'annual' and 'subsequent' things? I guess the R in MTR is ratio? Not clear. But note that MTR stands for Mean Tidal Range in usual tidal studies. Anyway I found these experiments at lines 180-204 somewhat unconvincing, although I do understand why you felt the need to inject some rigour into the choice of boundary values between Types. But the experiments do not cover the whole space of possibilities for amplitude and phase lag of all constituents concerned. The main thing to me is Figure 6 which shows nicely how Fsm varies with x and y. Why don't you then just define the boundaries between Type 1 etc. in an ad hoc way, similar to the way as F is divided in an ad hoc way for 'semidiurnal' etc. After all, in the end all these form factors are just handy coarse descriptive subdivisions for the tide. Anyway lines 180-204 need rewriting - see my comment at line 173 also. It is just not clear what you are doing. Response: The "moderating influence of" has been replaced by "strong influence of". The whole section on the experiments, including Figure A1, has been deleted. As suggested, simplified ad hoc (and less repetitive) explanations for selecting the boundaries between the different E types has been used instead as follows: "Below we explain how we set boundaries between the different E types around ANZ, using our case study data, and as summarised in Fig. 6. Firstly, in any semi-diurnal tidal regime (F<0.25) anywhere in the world where the amplitude ratio $N_2/S_2 < 1$, spring-neap cycles will feature clearly in the tidal height records. Thus, the boundary separating Types 1 and 2 from Types 3 and 4 occurs at $N_2/S_2 = 1$. Type 1 and 2 areas of the ANZ coast are characterized by relatively larger S2 amplitudes (19-40 cm) than areas with stronger perigean-apogean influences (2-18 cm) (Table 1). Secondly, tidal regimes with stronger spring-neap signals include places where spring-neap cycles occur as consecutive fortnightly cycles of similar magnitude (Type 1 or 'spring-neap' type regimes), and places where spring-neap signals dominate but with noticeable variability in the magnitudes of consecutive cycles due to subordinate perigean-apogean influences (Type 2 or 'in- termediate, spring-neap' regimes). In ANZ the strongest spring-neap influence occurs in the Cook Strait to Kapiti area, where harmonic analysis revealed an amplitude ratio of $N_2/S_2 = 0.35$ and an E value of 0.79 (Table 1). Examining the shapes of tidal height plots showed that Kapiti had the only completely spring-neap dominated tidal envelope amongst the case study sites. Hence the boundary between Type 1 versus 2 was set as E = 0.8 for ANZ, just greater than that of Kapiti and below the next strongest spring-neap influenced site, Nelson, where E = 0.9 (Fig. 6). Lastly, to set a boundary between 'perigean-apogean' and 'intermediate, perigean-apogean dominant' regimes (i.e. Types 3 versus 4), we again examined tidal height plots to determine a boundary value of E = 1.15, between the 'intermediate, perigean-apogean dominated' type regime of Napier (E = 1.147) and the 'perigean-apogean' type regime of Kaikoura (E = 1.162) (Table A1; Fig. 6)."

- see below. mention the other red blob. Response: The following text has been added: "As shown in Fig. 1c, such regimes unusual internationally, also occurring in limited areas of the Cook Islands and northeast of Pitcairn Islands in the Southwest Pacific Ocean; in Alaska's Bristol Bay, Canada's Hudson Bay and offshore of the North Carolina to Virginia coast in North America; on the north coast of the Bahamas in Central America; and in the Gulf of Ob in Russia".

- if you agree then drop 'theoretical experiments' here. Response: Yes – all mention of them, and the experiments themselves, have been dropped.

- these three do not all operate at 'synodic anomalistic timescales'. Why not just '..three key constituents (M2, S2 and N2).' At this point it occurred to me that a similar exercise could be conducted for areas of predominantly diurnal (but a bit mixed) tides. Could you speculate in this Discussion which parts of the world could benefit that way? Response: 3 key constituents change made. Using the FES2014 model for a similar exercise, we explored predominantly diurnal tidal regimes and found a possible area, the Ross Sea, Antarctica, where there are extremely weak (F>15), very weak (F>8) and weak (F>5) M2 and S2 amplitudes along with the areas of M2>S2, S2>M2, N2>S2

and S2>N2 amplitudes. Thus, our approach to classifying monthly tidal patterns can be applied to the Ross Sea diurnal tide area, but it is not as simple as the application in this paper to semi-diurnal ANZ regimes.

- this isn't necessarily true. Figure 1c shows where S2 is small compared to M2. It doesn't necessarily follow that perigean influences dominate. Response: We checked the results for all 'red blob' areas and in all the N2 is at least five times greater than the S2.

- what is 'low-frequency coastal flooding'? Response: This unhelpful descriptor has been deleted.

Table A1. Line 1 - you don't show tidal ranges, this will be confusing for most people. What you show on the last line are ranges of amplitudes and phase lags in your data set. Also the 'ranges' shown are crazy for some as shown e.g. see 6-360 for K1. But 360 degrees is the same as 0 degrees! Line 2 - values. Also the header should mention you show Types. Say if the phase lags shown are in Greenwich Mean Time or local time? if Greenwich then they are usually denoted by G. Response: Corrections to caption and number values (360 –> 0) made as suggested, and line 2 deleted. Phase lag reference added and notation amended to G. Also the columns in this table have been re-ordered in line with the suggested re-ordering of the Table 2 (now Table 1) columns.

Figure A1. I don't understand the 'under conditions summarised in Table A1'. Surely all one needs to know is which stations were used for these 3 examples. Response: Since the experiments have been deleted, this figure has also now been removed.

- doesn't matter much but Figure 2 looks like a simple coastline map to me that one could make with GMT or Matlab, so where do the fancy 'map layers' come in? And with an undesirable national coordinate system to boot instead of lat/lon? Response: The coordinate system and map outline were supplied as CorelDraw map layers. The coordinate system has been amended to internationally understandable Lat/Long, and the Acknowledgement note has been simplified to 'map layer'.

Table 1. Line 1. The word 'interval' in tides refers to the times of high tide since passage of the moon. What you are showing here are not intervals but the periods of beating of the shown pairs of constituents. And personally I would abandon columns 3 and 4 - you are not writing a text book here - certainly drop column 3 (and in M2/S2 – drop 'axial'. M2/N2 - drop 'relative'. line 3 'during the siderial month' –> during a month). And I would drop the Note which doesn't add anything. Response: Taking on board these comments, we have completely deleted Table 1 (and also Tables 3 and 4 as per your comments below) and re-ordered the remaining table.

Table 2. I would have a column 1 showing Type. And I would move Example Sites to be a column 2. First line of that: Equilibrium Theory (no footnote and no Note – you have already mentioned Defant in the text). Response: Table columns re-ordered and changes made as suggested (note – now Table 1 as per change above).

Table 3 - I guess this does no harm but it just repeats what has been given in the text. I would drop it. Response: This table is now deleted – it was superfluous with clearer formatting of Table 2 (now Table 1).

Table 4 - I don't understand this table. It is tied up with mention of the experiments, see comments above. I would drop this table as well. Response: This table is now deleted, in line with removal of the experiments as commented on above.

Figure 1 - just a suggestion but perhaps all panels could be made the same size. You have (a) large but that is for the normal F which is not the subject of this paper and can be found in many text books. Also for this, and also for the other colour maps in Figs 3,4 etc. could you have an arrow on the max colour as you have points on the maps with values which are in overflow. As for Figure 1 (c), you should mention somewhere in the text where the other red blob is. Near Tahiti? line 4 of caption '.... monthly tidal envelope using criteria described in section 3.' Then for (c) see my comment for line 230. Response: All 3 maps are now the same size. The overflow issue has been eliminated from this figure. The red blobs in 1 c have now been described in the text more fully, as indicated above. For Fig. 1c, see reply above. The caption for 1 c has also been adjusted.

Figure 2 - please use conventional lat/lon and not a national coordinate system no-one else will understand. As mentioned above there are places in the text (e.g. Stewart Is.) not shown. '&' –> 'and'. When Figure 2 is first mentioned in the text there is no mention of the Type 1, 2 etc. So you have to returrn to this figure after you discuss Figures 6 and 7 and mention the Types in Fig.2, and then please also use the same colours for the Types here as in Fig 6. Response: All changes made as suggested.

Fig 3 - arrows needed on colour scales e.g. for the overflow top-left of 3(d). The contour annotation bottom right of 3(f) is messy, please thin out the annotations. Also drop 'Unit' in 'Unit mm'. line 1 of caption - 'Amplitudes for'. Drop 'horizontal'. Line 2 - drop 'derived and'. Drop 'database'. 'at a scale of' –> 'on a grid of' Response: The scale of 3f has been changed to eliminate the overflow issue. The contours have also been thinned out as recommended. 'Unit' has been deleted from the scale bar. All caption changes have been made as suggested.

Fig 4 - as mentioned I would drop the a's in the headers and captions. Arrows on colour scales. line 1 of caption - drop 'horizontal'. Line 2 - drop 'database'. 'at a scale of' –> 'on a grid of' Fig 5 - drop a's Response: The overflow area in Fig. 4c has been clearly delineated and labelled, and the suggested caption changes are made, including removal of 'a's.

Fig 6 - this is actually a useful plot. Use another colour instead of pink which is too much like red. drop a's. Use same colours for the Types as in Fig 2. Add dotted or dashed lines also for the Fsm boundary values chosen to define Types 1-4. Also what would be useful also would be to have values from FES2014 for the whole NZ coastline - that might be a fiddly computing exercise but is obviously possible. Response: Color, dashed line boundary, and 'a' changes made as suggested. We decided not to made a

FES2014 model based version of this diagram for the whole ANZ coast due to our focus on observational data in this diagram (including the Walters et al. data comparison) and also since we use the FES2014 data to do this task (albeit in map rather than plot form), classifying the whole ANZ coast into E type categories, in our newly expanded Figure 7.

Fig 7 - overflow arrow. could roughly the same colours be used as for Fig 6 as far as possible? That has red-green-blue-pink for types 1-4 whereas this has green-yellowred more or less (the blue is not used). line 2 - .. see Figure 5 for definitions and examples of .. Response: All colour and caption changes made as suggested. Figures 2, 5, 6 and 7 now have the same colours for all E types.

References Allen J., Davis C., Giovinazzi S., Hart DE.: Geotechnical and flooding reconnaissance of the 2014 March flood event post 2010-2011 Canterbury Earthquake Sequence, New Zealand, Report No. GEER035, commissioned by the Geotechnical Extreme Events Reconnaissance Association, 134 pp, http://dx.doi.org/10.18118/G6001Z, 2014. D'Onofrio, E. E., Fiore, M. M., and Romero, S. I.: Return periods of extreme water levels estimated for some vulnerable areas of Buenos Aires, Cont. Shelf Res., 19(13), 1681-1693, 1999. Hart, D. E., Giovinazzi, S., Byun, D.-S., Davis, C., Ko, S.-Y., Gomez, C., Hawke, K., and Todd, D.: Enhancing resilience by altering our approach to earthquake and flooding assessment: multi-hazards, 16th European Conference on Earthquake Engineering, 18 to 21 Jun, 2018, Thessaloniki, (12164) 13 pp, 2018. NIWA, National Institute of Water and Atmospheric Research.: Tide Forecaster, accessed 28 December 2019 from: https://tides.niwa.co.nz/, 2019.
* * *
[Figure]

**Figure R1. (Top) One month of tidal heights for Christchurch on the South Island's east coast (left) versus Westport on the South**
**Island's west coast (right); and (Below) 2020 predicted tidal 'red alert' days for Christchurch (left) and the South Island West Coast**
**(right), Aotearoa New Zealand (generated using NIWA, 2019).**

[Figure]

**Figure 6. Plot of the relationship between the $\frac{N_2}{S_2}$ and $\frac{S_2}{M_2}$ amplitude ratios (y and x axes respectively) versus $E$ values (shown as plot**
**contours), with data points corresponding to Aotearoa New Zealand waters Type 1 sites (red dots); Type 2 sites (green dots); Type**
**3 sites (blue dots); and Type 4 sites (yellow dots), all from Table A1; and tidal data representative of the greater Cook Strait area**
**(grey crosses) from Walters et al. (2010, Tables 1 and 3).**

**Fig. 2.** Figure 6. Plot of the relationship between the N_2/S_2 and S_2/M_2 amplitude ratios (y and x axes respectively) versus E values (shown as plot contours), with data points corresponding to Aotearoa N

[Figure]

[Figure]

**Fig. 3.** Figure 7. Distribution of monthly tidal envelope factor (E) values (a); and types (b); in the waters around Aotearoa New Zealand, including in the Cook Strait area between the two main islands (c); ca

---

## Referee Comment (RC2) · Anonymous Referee #2 · 5 Feb 2020

The paper is basically acceptable, and Figures 1b, 1c, and 6 are useful. Most of the paper is devoted to trying to find the numerical delineations between spring-neap and perigean regimes, and that is a little tedious, as the boundaries are bound to be fuzzy and perhaps not applicable everywhere, even in purely semidiurnal regimes. (For example, the moderating role of K2, which likely causes variations throughout the year, isn't brought up. This, however, isn't fatal, since this whole exercise is merely to produce rough rules of thumb.)

I didn't spot anything that is clearly in error, just minor issues, listed below. Some of these issues involve odd, almost off-the-cuff remarks in the introductory material rather

than in the technical material.

Numbers below refer to Line Numbers in the paper.

24 - Neither the Egbert nor Stammer papers have anything to do with sea level change or gravimetry.

41 (also Table 1): Is it a sidereal month or a tropical month?

47: "Far less attention" - There is a good reason for that, as the major tides are obviously most important for prediction. And why specify "modern" in this context? It's always been the case.

216 "having common ways of describing different types of tidal envelope is essential for living safely and productively..." – ESSENTIAL, really? That seems overblown. In fact, I consider a full-up tide prediction to be far more essential.

Along the same lines, is it really necessary to have similar statements in the Abstract? The first and last sentences of the Abstract seem to me to be quite a stretch in trying to justify the work.

60: what plates NZ sits on is rather irrelevant to the subject.

88: I'm not sure why "sidereal" is used in reference to K1 and O1. "Declinational" or just "diurnal" seems more apt.

173: "moderating" is an odd way to refer to M2.

Table A1. It should state these are Greenwich phase lags (which I believe to be the case), since lower-case "g" is often used to denote a local phase. One could also argue that the F value based on "Equilibrium Theory" ought to be a function of latitude.

Is Table 4 really necessary? Aren't Tables 2 and 3 and Figure 6 sufficient?

And finally a point on names. Presumably the government of New Zealand has not (yet?) changed the country name to Aotearoa. Is there a reason to use (what I assume

is) Maori throughout this paper – including even for the Pacific Ocean and Tasman Sea? I suspect that indigenous Australians have a different name for these. Why not use those? Why not use Korean as well? I don't really see the point of using an obscure indigenous name for the Pacific Ocean.

---

## Author Comment (AC2) · 6 Feb 2020

Thank you for these comments. Here we have copied each individual reviewer comment, and written below it a response.

Place names: There is some inconsistency in the place names used in the paper. For example, Cook Strait is referred to as 'Cook Strait' (line 73 and others) and 'Te Moanao-Raukawa Cook Strait' (line 70 and Figure 2). Lines 60 and 61 give both alternative names for the North Island and South Island. The English and Māri names are alternatives; it is not necessary to use both - choose one form and use it consistently. The official name for Stewart Island is 'Stewart Island / Rakiura', not 'Rakiura or

[Figure]

Stewart Island' as shown in line 61. It is recommended that place names used are as shown in the NZ Place Names Gazetteer. Cook Strait is just 'Cook Strait' (not an official names but a recorded one), 'Castle Point' (line 117) is 'Castlepoint' (official). 'Aotearoa New Zealand' has been used for the name of the country (and abbreviated to ANZ) but until an Act of Parliament is passed the country is 'New Zealand'. Response: The approach taken was: at first mention of a name we used use both Te Reo Maori and English names, thereafter referring to each place by which of these two is the most commonly used name today. This was combined with including both languages on the map. The reason for including both official written language names at first mention/ on the map was to recognise, with equivalence, both types of official place name. This was also pragmatic, to try to give our paper some time-proofing, since it is not uncommon today for place names in our country to revert officially from their English to their Maori version. By including both, we thought our paper might withstand such changes and still be readable in the future. However in recognition of the direction to use only one form for each place name, and recognising that Copernicus has an international audience, we have selected one name for each place and used that consistently. By contrast, the case of the Castlepoint typo (Castle Point on line 117) was an error and we have fixed this now.

Line 82: It would be helpful to point out that the results of the analysis of the records from the additional 33 locations are presented in Figure 6, and a sentence or two summarising those results would be appropriate. Response: Yes, our text now reads: "The Cook Strait's tides were explored in detail by Walters et al. (2010): our Fig. 6 includes a re-analysis of their data using the E ratios. Note that the Cook Strait data includes 4 sites in the Type 1 category, as well as a number of Type 2 and Type 4 sites, and one Type 3 site, revealing this small strait to be a concentrated area of monthly tidal envelope diverse".

Line 83: What does 'reach the strongest' mean? Response: Reach has been changed to "are" as follows: "where spring-neap tides are the strongest in the country".

Line 93 and 94: The text here states that Figures 3 and 4 map the constituent amplitudes and ratios listed in Table 1, but surely the figures are derived from the FES2014 model as stated in the captions for Figures 3 and 4. Response: Lines 93 to 94 had stated: "In order to better understand the key constituents responsible for shaping tidal height forms around ANZ, we first mapped variability in the amplitudes of the semi-diurnal and diurnal constituents listed in Table 1 (Figure 3) and of the ratio values of the semi-diurnal constituent amplitudes (Figure 4)". That was meant to indicate that we used the list of constitutes named in Table 1 (i.e. those involved in spring-neap cycles, and in perigean-apogean cycles, plus the diurnal tides - i.e. M2, S2, N2, K1, O1) to determine what to make plots of using the FES2014 data. Table 1 has now been deleted from our paper, according to the Woodworth review suggestion, so any confusion created by our reference to this table is now deleted.

Line 117: Castlepoint is on the east coast of the North Island. Response: Thank you for picking up this typo – "South" has been corrected to "North".

Line 125: What does 'combined variability' mean? The rest of this sentence is difficult to follow - a diagram might help? Response: We have deleted "combined" and altered this paragraph so that it now reads: "We distinguished these two envelope types via the tides generated by variability in the amplitude ratios of $S_2/M_2$ and $N_2/M_2$ (i.e. of the spring-neap cycle, and perigean-apogean cycle, forming tides, respectively). In brief, the $S_2/M_2$ and $N_2/S_2$ amplitude ratios vary widely around NZ, with highest values in the west, lowest values in the east, and intermediate values to the north and south (Fig. 4). By comparison, the $N_2/M_2$ amplitude ratios are relatively stable and high, except in a relatively small area of Cook Strait to the Kapiti coast, where this ratio drops and thus spring-neap cycles predominate (see 'spring-neap' Type 1 regimes above). The variability in these two ratios means that, except where we find 'spring-neap' or 'perigean-apogean' monthly tidal envelopes types, spring-neap tides do occur but the overall monthly envelope shape is fundamentally altered (asymmetrically) due to the perigean-apogean influence".
Figure 2: Castlepoint is shown out of its true position. Response: Thank you – the map
has been adjusted.

Key
- ● LINZ site
- ■ NIWA site
- ● Type 1
- ● Type 2
- ● Type 3
- ● Type 4

North Cape, Korotiti Bay, Auckland, Onehunga, Manukau, Tauranga, Moturiki Island, Lottin Point, Taranaki, North Island, Gisborne, Napier, Wellington, Kapiti, Castlepoint, Westport, Nelson, Charleston, Cook Strait, Kaikoura, Sumner, Pacific Ocean, South Island, Port Chalmers, Green Island, Puysegur, Bluff, Dog Island, Stewart Island/Rakiura, Tasman Sea

Raoul Island
- Fishing Rock
- Boat Cove
- 29°16' S
- 177°55' E

Chatham Islands
- Owenga
- 44° S
- 176°30' W

**Fig. 1.** Figure 2

---

## Author Comment (AC3) · 5 Mar 2020

Reply to Anonymous Referee #2's Interactive comment on "A monthly tidal envelope classification approach for semi-diurnal regimes with variability in S2 and N2 tidal amplitude ratios" by Do-Seong Byun and Deirdre E. Hart, Received and published: 5 February 2020

Thank you for these helpful comments. Here we have copied each individual comment, and written below it a response.

Reviewer introduction: The paper is basically acceptable, and Figures 1b, 1c, and 6

are useful. Most of the paper is devoted to trying to find the numerical delineations between spring-neap and perigean regimes, and that is a little tedious, as the boundaries are bound to be fuzzy and perhaps not applicable everywhere, even in purely semidiurnal regimes. (For example, the moderating role of K2, which likely causes variations throughout the year, isn't brought up. This, however, isn't fatal, since this whole exercise is merely to produce rough rules of thumb.) I didn't spot anything that is clearly in error, just minor issues, listed below. Some of these issues involve odd, almost off-thecuff remarks in the introductory material rather than in the technical material. Numbers below refer to Line Numbers in the paper. Response: Thank you for taking the time to review our paper. We feel we have been able to very significantly improve our paper based on the 3 sets of feedback provided. Please see below specific responses to this review.

24 - Neither the Egbert nor Stammer papers have anything to do with sea level change or gravimetry. Response: The Egbert et al. (1994) reference has been deleted from here (details in our response to the Woodworth review). Stammer et al. (2014, p243) state as a justification for their accuracy assessment paper that "An especially important application for accurate tide models is providing tide "corrections" to various measurements so that smaller nontidal signals may be studied. For example, barotropic tide models are used regularly to remove tidal variability from space geodetic observations; this is a critical necessity for successful satellite altimetry [e.g., Fu and Cazenave, 2001] and satellite gravimetry [Seeber, 2003; Visser et al., 2010], and in both cases improved tidal corrections lead to a reduction of aliased tidal "noise" in nontidal signals of interest". It is this point from Stammer et al. (2014) that we wished to point our readers to. Our revised text now makes this clearer by the repositioning of this reference as follows: "An understanding of tidal water level variations is fundamental to.... accurately resolving non-tidal signals of global interest (Stammer et al., 2014), such as in studies of sea level change".

41 (also Table 1): Is it a sidereal month or a tropical month? Response: This table has
been deleted (see response to Woodworth review).

47: "Far less attention" - There is a good reason for that, as the major tides are obviously most important for prediction. And why specify "modern" in this context? It's always been the case. Response: Yes, thank you for your comment - we have removed this text and recalibrated the tone of remaining text. The revised paragraph now reads: "Tidal envelopes at monthly scales depend on tidal regime. In general, semi-diurnal tidal regimes often feature two spring-neap tidal cycles per synodic (lunar) month. These two spring-neap tidal cycles are usually of unequal magnitude, due to the effect of the moon's perigee and apogee, which cycle over the period of the anomalistic month. In contrast, diurnal tidal regimes exhibit two pseudo spring-neap tides per sidereal month. For semi-diurnal regions where the N2 constituent contributes significantly to tidal ranges, tidal envelope classification should consider relationships between the M2, S2, and N2 amplitudes. The waters around NZ represent one such region: here the daily tidal form is consistently semi-diurnal, but large differences occur between sites within this region in terms of their typical tidal envelope types over fortnightly to monthly timescales. More than eighty years after the development of the ever-useful daily tidal form factors, attention to the regional distinction between different tidal envelope types within the semi-diurnal category forms the motivation for this paper".

216 "having common ways of describing different types of tidal envelope is essential for living safely and productively..." – ESSENTIAL, really? That seems overblown. In fact, I consider a full-up tide prediction to be far more essential. Along the same lines, is it really necessary to have similar statements in the Abstract? The first and last sentences of the Abstract seem to me to be quite a stretch in trying to justify the work. Response: We have deleted 'essential' from our text here and replaced it with 'helpful'. The term in our revised abstract reads 'of use', which is a much milder claim than previously written. The abstract has been much modified in response to a similar point made in the Woodworth review, also improving it in terms of the comment made here.
60: what plates NZ sits on is rather irrelevant to the subject. Response: We agree that the names of the plates are not essential to our core paper topic. However they are useful in the context of explaining the long narrow shape of the chain of islands that make up New Zealand, and this shape plays a role in interacting with our ocean tides. The revised text now reads: "New Zealand (Fig. 2) is a long (1600 km), narrow ( $\leq$ 400 km) country situated in the south-western Pacific Ocean and straddling the boundary between the Indo-Australian and Pacific plates. Its three main islands, the North Island, the South Island, and Stewart Island/ Rakiura, span a latitudinal range from about 34° to 47° South".

88: I'm not sure why "sidereal" is used in reference to K1 and O1. "Declinational" or just "diurnal" seems more apt. Response: Yes thank you - we have replaced 'sidereal' with 'diurnal' in our revised text.

173: "moderating" is an odd way to refer to M2. Table A1. It should state these are Greenwich phase lags (which I believe to be the case), since lower-case "g" is often used to denote a local phase. One could also argue that the F value based on "Equilibrium Theory" ought to be a function of latitude. Response: Thank you for these comments – we have addressed all of them. Table A1 now has correct reference to Greenwich phase lags in the caption and the corrected capital G parameter label in the table proper. Regards line 173, we removed the sentence with "moderating" in it (see details in the reply to the Woodworth review). The text now reads: "We distinguished these two envelope types via the tides generated by variability in the amplitude ratios of S\_2/M\_2 and N\_2/M\_2 (i.e. of the spring-neap cycle, and perigean-apogean cycle, forming tides, respectively). In brief, the S\_2/M\_2 and N\_2/S\_2 amplitude ratios vary widely around NZ, with highest values in the west, lowest values in the east, and intermediate values to the north and south (Fig. 4)".

Is Table 4 really necessary? Aren't Tables 2 and 3 and Figure 6 sufficient? Response: Thank you and yes it was unnecessary so we have deleted this table (see response to Woodworth review where we expand on our table deletions and adjustments). Basically

OSD
only a revised version of the original Table 2 remains (now re-labelled Table 1), with the original Tables 1, 3 and 4 deleted since they were unnecessary.

And finally a point on names. Presumably the government of New Zealand has not (yet?) changed the country name to Aotearoa. Is there a reason to use (what I assume C2 OSD Interactive comment Printer-friendly version Discussion paper is) Maori throughout this paper – including even for the Pacific Ocean and Tasman Sea? I suspect that indigenous Australians have a different name for these. Why not use those? Why not use Korean as well? I don't really see the point of using an obscure indigenous name for the Pacific Ocean. Response: Recognising that Copernicus has an international audience we have selected only one name for each place within the country now, called the country 'New Zealand', and used English ocean names consistently throughout the paper and in a revised version of Figure 2 (see response to Rowe comment for the new figure and more detail of these changes).

**OSD**

---

## Editor Comment (EC1) · Mattias Green (Editor) · 14 Apr 2020

The authors have provided a strong reply to the reviewers' comments, and I feel this is close to publication. I have asked for Rev 1 to look at it again since they asked for major revisions, to ensure all comments have been taken into account.

I have one minor request of the authors for the final version: please don't use the rainbow colour scheem in the figures - they are really tricky for people with colour blindness to see. Please look at the cmocean pacakge and use the colormaps there instead.

---

## Author Response (AR1)

**Response to reviews and comments on the paper* "A monthly tidal envelope classification approach for semi-diurnal regimes with variability in $S_2$ and $N_2$ tidal amplitude ratios"**

Do-Seong Byun[1], Deirdre E. Hart[2]

[1]Ocean Research Division, Korea Hydrographic and Oceanographic Agency, Busan 49111, Republic of Korea
[2]School of Earth and Environment, University of Canterbury, Christchurch 8140, Aotearoa New Zealand

*Correspondence to*: Deirdre E. Hart (deirdre.hart@canterbury.ac.nz)

**Introduction** We are very grateful for the reviews and comments received on this paper as collectively they has been very useful in helping to improve the paper. Below we have copied each individual reviewer comment, written below it a response in blue font, and then copied any insertions or modifications to the text. Almost all suggested changes have been adopted wholesale, but discussion and a couple of the points remain below regarding the link made to flooding hazard. Following our responses and changes made sections, we include in this file a revised version of the full paper with track changes. We will submit the revised paper file separately as well, according to the Copernicus instructions.

**1. Reply to *Interactive comment by Philip Woodworth, received and published 22 Dec. 2019**

*Abstract - if I had written this abstract I would have used the useful words on lines 206-214 of the Discussion. For example, I can see that the new form factor could inform about shoreline ecology as ecology depends on the tidal climatology. However, I cannot see that it is much use in discussion of inundation hazards and climate change; for that one would be interested primarily in the character of sea level extremes and not just on simple descriptions of the tide.*

- **Response:** Thank you for this suggestion regards the discussion text – we have used some of this text to replace the original opening sentence of the abstract. Regards the inundation hazards and climate change link here with perigean-spring tides, this comment relates our experiences in Christchurch (e.g. Allen et al., 2014; Hart et al., 2018). This city (marked by the Sumner gauge site in Fig. 2) experienced up to 1 m relative sea level rise in coastal and river proximal suburbs due to subsidence during the Canterbury Earthquake Sequence (CES, 2010-2011). This instantaneous sea level change was equivalent in magnitude to that which had been predicted (in absolute as opposed to relative terms) for the next 50 to 100 years due to anthropogenic climate change and accelerated sea level rise. We thus use Christchurch as a 'laboratory' to consider what 1 m of sea level rise might look and feel like in a delta city (of which there are many similar settings in seismically active areas worldwide), albeit with process-response timescales being rather different to those under climate change scenarios. One of the greatest effects has been enhanced flooding issues, much more so than other coastline hazards such as erosion. Since the city relies on river and estuary drainage conduits, in particular, when pronounced perigean-spring tides occur in combination with sustained rainfall events, inland riverside and low-lying coastal suburb flooding is widespread, deep and persistent. It would seem that around half of the city had little freeboard, and that buffer has been significantly reduced with the CES such that monthly high tides pose issues for the lowest lying areas nowadays. The backwater effects of high tides combine with atmospheric low pressure and sustained precipitation events to extend the reach of flooding. We suspect that the latter will continue to be enhanced as the baseline mean sea level rises with climate change, meaning less ability to cope with perigean-spring tides. Since the CES, high tide alerts have become of wider public interest as they are now commonly associated with flooding. Understanding the frequency of such tidal alert days has been of use to those at the coal face of flooding, in terms of emergency responses, as well as in making decisions about whether to stay or retreat from subsidence affected areas. We therefore see monthly tidal height patterns as intricately linked to questions of initial sea level rise effects in our city.

•   **The altered text now reads:** "**Abstract.** Daily tidal water level variations are a key control on shore ecology; access
to marine environments via boat and shipping infrastructure such as ports, jetties and wharves; drainage links between
the ocean and coastal hydrosystems such as lagoons and estuaries; and the duration and frequency of opportunities to
access the intertidal zone for recreation and food harvesting purposes. Further, high perigean-spring tides interact
with extreme weather events to produce significant coastal inundation in low-lying coastal settlements such as on
deltas. Thus an understanding of daily through to monthly tidal envelope characteristics is fundamental to resilient
coastal management and development practices".
***12 -*** *remove 'database'. 'theoretical experiments' –> 'theoretical arguments' maybe.*
•   **Response:** Both 'database' and 'theoretical experiments' have been removed (see also this reviewer's comment on
paper line 139 below, and our response).
•   **The altered text now reads:** "Analyses of tidal records from 27 stations are used alongside data from the FES2014
tide model in order to find the key characteristics and constituent ratios of tides that can be used to classify monthly
tidal envelopes".
***14 -*** *the symbol Fsm is a clunky one and even impossible to write on an ascii keyboard. What is it supposed to mean? A form*
*factor showing S2's influence on M2? But what about N2 i.e. Fnm? I would have invented a simpler symbol such as F-prime*
*or maybe E for envelope?*
•   **Response:** As suggested, we have changed $F_M^S$ throughout the paper to the much simpler notation of $E$.
***20 -*** *I don't see that the first two references are really relevant to this sentence. Cartwright is a history of tidal science.*
*D'Onofrio discusses Buenos Aires only and not spatial variation. The Nicholls reference is ok.*
•   **Response:** The first two references have been removed from the sentence: "*Successful human-coast interactions in*
*the world's low-lying areas are predicated upon understanding the temporal and spatial variability of sea levels*".
Please note that we meant the phrase 'temporal and spatial variability in sea levels' to encompass a wide range of
processes including cyclical tidal height variations, and were not meaning mean sea level variations alone, a topic
best highlighted using the third reference. We have added the reference Woodworth et al. (2019) to emphasise this
wider meaning.
•   **The altered text now reads:** "Successful human-coast interactions in the world's low-lying areas are predicated upon
understanding the temporal and spatial variability of sea levels (Nicholls et al., 2007; Woodworth et al., 2019)".
***24 -*** *'and gravimetry'. What does that refer to? Space gravimetry by missions such as GRACE? I would drop that. Then again*
*the references are apparently random – Egbert et al. describes one particular model, while Stammer et al. describes many*
*including Egbert. So why is Egbert here and not all the others?*
•   **Response:** We have removed "*and gravimetry*" and the Egbert et al. (1994) reference. Stammer et al. (2014, p243)
stated the point, which we repeated in shorter form, that "*An especially important application for accurate tide models*
*is providing tide "corrections" to various measurements so that smaller nontidal signals may be studied. For*
*example, barotropic tide models are used regularly to remove tidal variability from space geodetic observations; this*
*is a critical necessity for successful satellite altimetry [e.g., Fu and Cazenave, 2001] and satellite gravimetry [Seeber,*
*2003; Visser et al., 2010], and in both cases improved tidal corrections lead to a reduction of aliased tidal "noise"*
*in nontidal signals of interest*". We have made our use of the Stammer et al. point clearer by repositioning the
reference, as below.
•   **The altered text now reads:** "An understanding of tidal water level variations is fundamental to resilient
inundation management and coastal development practices in such places (Cartwright, 1999; Masselink et al., 2014;
Olson, 2012; Pugh, 1996), as well as to accurately resolving non-tidal signals of global interest (Stammer et al.,
2014), such as in studies of sea level change".
***26 -*** *I would have the equation here i.e. F=(K1+O1)/(M2+S2) and not just words, like your equation (1) below which would*
*become (2)*
•   **Response:** We have added this equation explicitly, as suggested, and renumbered the other equation.

• **The altered text now reads:** "Originally developed by van der Stok (1897) based on three regime types, with a fourth
type added by Courtier (1938), this simple and useful daily form factor comprises the ratio between the combined $K_1$
and $O_1$ diurnal amplitudes versus the combined $M_2$ and $S_2$ semi-diurnal amplitudes via the equation:

$$F = \frac{K_1 + O_1}{M_2 + S_2}$$ (1)".

**26-27 -** *if you have four you can't add a fourth?*
• **Response:** This has been clarified in the text. Originally van der Stok (1897) divided tidal regimes into three types
using the $F$ equation, while Courtier (1938) added a fourth (daily) tidal regime type.
• **For altered text:** please refer to the altered text in the response immediately above this one.

**28 -** *aren't they the same form factor (singular)?*
• **Response:** This has been corrected in the text.

*I am not familiar with the van der Stok and Courtier references which are very old and I don't think many other readers will*
*be either. How did you come across them? If in a more recent history of tides or a text book on tides then please add that.*
• **Response:** Van der Stok (1897) was available to us via interlibrary loan. We borrowed an original 1897 large format
book from California through the UC library – see available copies here: https://www.worldcat.org/title/wind-and-
weather-currents-tides-and-tidal-streams-in-the-east-indian-archipelago/oclc/488220907. It is an interesting piece of
work as it clearly outlines the $F$ equation and three of the four tidal regime types in common usage today, in a work
dating back over 120 years. We feel that it is best to leave this reference in our paper discussion of the origins and
history of use of $F$, not least to give credit to this early author. We found out about van der Stok's work from its
citation in Courtier (1938), which is available online, in a PDF English translation, from:
https://journals.lib.unb.ca/index.php/ihr/article/download/27428/1882520184. We have added this web link in our
reference list entry for Courtier (1938) to make it more accessible to readers. Both of these references were located
via a Google search (in contrast our university library multi-search returned no useful results).
• **The altered text now reads:** "Courtier, A.: Marées. Service Hydrographique de la Marine, Paris (English translation
available from: https://journals.lib.unb.ca/index.php/ihr/article/download/27428/1882520184), 1938".

**34-36** *this is a garbled sentence. Could you please reword?*
• **Response:** We have reworded this sentence into two shorter, clearer sentences as follows.
• **The altered text now reads:** "The daily tidal form factor identifies the typical number (1 or 2) and form (equal or
unequal tidal ranges) of tidal cycles within a lunar day (i.e. 24 hours and 48 minutes) at a particular site. In contrast,
the term 'tidal envelope' describes a smooth curve outlining the extremes (maxima and minima) of the oscillating
daily tidal cycles occurring at a particular site through a specified time period".

**45-47** *This isn't right. You say yourself that NZ tides are unusual so the reviews of Andersen etc. cannot be blamed for focusing*
*on the main constituents relevant to global studies. However, that does not mean those authors were disinterested in other*
*constitents. In fact one main aim of such studies was to determine how well the total tide could be determined which*
*necessitates accuracy in N2 etc.*
• **Response:** We have deleted these lines.

**56 -** *as mentioned above I can't see form factors (of whatever kind) being directly relevent to coastal flooding hazards work,*
*but if I am wrong please give references.*
• **Response:** You are absolutely reasonable to question the unusual link drawn in our paper between form factors and
coastal flooding hazards work. However, we would like here to offer explanation for why we think this link exists for
some places (delta cities) and is relevant. Again using Christchurch as an example - this city is situated towards the
centre of NZ's east coast region of strongly perigean-apogean influenced tides. The city is constructed (like Tokyo,
Jakarta, Charleston NC, and many other delta cities) on a low-lying, formerly swampy, coastal progradation and river
delta plain in a seismically active area. This physical setting, combined with imprudent development, has influenced the flood hazard: major flooding occurs when periods of sustained heavy (as opposed to high intensity) rainfall
produce river and overland flows which fail to drain efficiently through the city's distributed, gravity-based and sea
level connected drainage network (Hart et al., 2018). One of the key factors that determines whether or not a sustained
rainfall event will result in widespread and severe flooding, or not, is the tides. Flooding is more likely during
perigean tides, since these times feature periods of more than a week with particularly high tidal ranges. As illustrated
in the Fig. R1 (top), unlike in spring-neap dominated areas, periods of high tidal ranges in Christchurch can last for
well over a week with short duration periods of smaller range tides between, when flooding is less likely. This means
that high tide 'red alert' days (Fig. R1 (below)) can last for more than a week, and there is an increased chance that
these might coincide with sustained rainfall events, than in more spring-neap dominated regions which feature the
distinct and regular punctuations of the lower range neap tides. This is a subtle but genuine reason why we believe it
is important for 'delta cities' like Christchurch to consider their monthly tidal pattern when considering the multiple
factors that influence flooding. An additional aspect of this idea relates to how we quantify future flood risks and
return periods under changing climate (not to mention in the multi-hazard context of future seismic activity, e.g. Allen
et al., 2014). The tidal height patterns will not be hugely influenced by climate change, so we can already produce
accurate frequency histograms and probability distributions for future high tide levels, like that conducted by
D'Onofrio et al. (1999) for Buenos Aries. Future rainfall and storm surge statistics are harder to predict under
changing climate and need to be combined with the more predictable tidal water level contributions to establish
accurate flooding and inundation risk predictions. In the past we in NZ focussed on flood return periods established
using historical water level records, but this is no longer a robust practice since the more predictable tidal water level
probabilities need to be combined with the changing atmospheric components to produce altered flood risk estimates
for the future. All this is in a relatively newly colonised country where hydrological data records are short. Our point
is partly that amongst all this uncertainty, at least the tidal pattern component of these hazards is nicely predictable,
so we encourage colleagues to take tidal patterns into account in their flood hazard analyses (something that has been
lacking in past flood analyses). We hope to make the case for the connection between tidal envelope pattern and flood
hazards in an upcoming ASCE (2020) monograph paper on flooding and inundation multi-hazards, but realise that
this idea is only hinted at in our current paper. Please do recommend if we should delete text making this link in our
current paper, or if we should leave it in, albeit as a fleeting mention, or some other suggestion.

[Figure]

Figure R1. (Top) One month of tidal heights for Christchurch on the South Island's east coast (left) versus Westport on the South Island's
west coast (right); and (Below) 2020 predicted tidal 'red alert' days for Christchurch (left) and the South Island West Coast (right), New
Zealand (generated using NIWA, 2019).

*61 - why don't you just have a simple map here for the reader to refer to i.e. Figure 2, and not wording such as 'latitudinal*
*gradient' - you mean range of latitude. Having the Type information in the figure is ok but you have to return to that later (see*
*below about that)*

• **Response:** We have corrected the word 'gradient' to 'range' and added a citation pointing the reader to Fig. 2 at this
point in the text, as suggested. Also, in Results section 3.1 we have added another mention of Fig. 2, highlighting at
this stage the observation station colour coding of their identified monthly tidal envelope types.

• **The Section 2.1 altered text now reads:** "New Zealand (Fig. 2) is a long (1600 km), narrow (≤400 km) country
situated in the south-western Pacific Ocean and straddling the boundary between the Indo-Australian and Pacific
plates. Its three main islands, the North Island, the South Island, and Stewart Island/ Rakiura, span a latitudinal range
from about 34° to 47° South".

• **The Section 3.1 altered text now reads:** "Figure 5 illustrates the four types of monthly tidal envelope found around
NZ as idealized types, two with stronger spring-neap signals (Types 1 and 2, see Fig. 5 a-b) and two with stronger
fortnightly perigean-apogean signals (Types 3 and 4, see Fig. 5 c-d) while Fig. 2 includes a colour coded classification
of the observation stations into the four tidal envelope types".

*62 - what are 'absolute tides'?*

• **Response:** We have amended the sentence to read "…*micro through to macro tidal ranges*".

***68-69 -*** *this business of a pair of amphidromes to the NW and SE is not easy for the reader to appreciate from your wording*
*alone, and the amphidromes are in fact a long way NW and SE and off the maps of Figure 3. So you have to point the reader*
*to where he can see a map of M2 in the SW Pacific - ideally a map from FES2014 as you have focused on that. Or see Fig 5.1*
*of Pugh and Woodworth (2014) which was provided by Richard Ray - i.e. a wider area than you have used for Fig.3. Anyway*
*I don't think it is right to say S2 has a single wavefront and amphidrome in the SE. Take a look at Figure 4 of Walters et al.*
*(2001) and you will see a pair of them close together in the SE. And I would drop mention of the Coriolis effect and simply say*
*that they rotate anticlockwise.*
• **Response:** Regards the description of the $M_2$, we have added a citation Pugh and Woodworth (2014) Fig. 5.1 – thank
you for this suggestion. We have amended our description of the $S_2$ and $K_1$ tide amphidromes in line with Walters et
al. (2001; 2010). Mention of the Coriolis Force has been removed as suggested.
• **The altered text now reads:** "Highly complex tidal propagation patterns occur around NZ, including a complete
semi-diurnal tide rotation, with tides generally circulating around the country in an anti-clockwise direction. This
occurs due to the forcing of $M_2$ and $N_2$ tides by their respective amphidromes, situated northwest and southeast of the
country respectively, producing trapped Kelvin waves (for a map of the $K_1$ and $M_2$ amphidromes see Fig. 5.1 in Pugh
and Woodworth, 2014). The $S_2$ and $K_1$ tides propagate northeast to southwest around NZ. This results in a southward
travelling Kelvin wave along the west coast, and small $S_2$ and $K_1$ amplitudes along the east coast, with amphidromes
occurring southeast of New Zealand (Walters et al. 2001; 2010)".
***77 -*** *'years' is misleading as it suggests you have used many years per station whereas Table A1 shows you used only one year*
*for each. Have the amplitudes and phase lags in Table A1 been adjusted for nodal variations according to equilibrium*
*relationships? Or are they the observed amplitudes for the years shown? See below for other comments on this table. I would*
*have prioritised the FES2014 model over the tide gauge data as the main aspects of what you are trying to show are best done*
*with the model. Then at the end of the paper you can show your findings from FES2014 are consistent with those from the tide*
*gauge positions.*
***80 -*** *you mean 'in comparison with values obtained from the tidal potential or Equilibrium Tide'*
• **Response to 77:** We have added the adjective "individual" to highlight that an individual year was analysed for each
of the 27 observation stations. In our paper we have chosen to initially employ the analysis of observational data, later
on using the FES2014 model data to check our findings and extend our spatial coverage. Although the observational
data does not have the same uniform coverage of the FES214 data, we prefer to initially use the observational data
since it represents real in situ records that are accurate at the coast, while the FES2014 data is helpful for our final
classification around the whole country (i.e. Figure 7).
• **Response to 80:** Yes, thank you – we have used your suggested (clearer) wording.
• **The altered text now reads:** "For both the LINZ and NIWA data, an individual year of good quality hourly data was
selected for analysis per site from amongst the multi-year records. The 27 individual year sea level records were then
harmonically analyzed using T_Tide  (Pawlowicz et al., 2002) with the nodal modulation correction option, to
examine spatial variation in the main tidal constituents' amplitudes, phase-lags, and amplitude ratios between regions
(see Table A1 for raw results) and to compare them with values obtained from the tidal potential or Equilibrium
Tide".
***81 -*** *'amplitude data' –> 'amplitudes'. 'was sourced' –> 'was obtained'*
***82 -*** *days' length or days in length*
***83 -*** *tides are the strongest*
***85, 242 and 268 -*** *Carrere has an accent over the first e*
***86 -*** *dataset –> model. experimental plots –> studies (maybe)*
***88 -*** *siderial –> diurnal.*
***92-93 -*** *mapped spatial variability*
• **Response:** All changes made exactly as suggested. The accent has been added everywhere that Carrère's work is
cited.

**94-95** - *.. those from the Equilibrium Tide (Defant, 1958). It was not Defant's theory. Anyway you might better refer to*
*Cartwight and Tayler (1971) for example.*
• **Response:** We have altered the misplaced reference to Defant (1958) to indicate that this was the source from which
we obtained the Equilibrium Tide data, and not the source of Newton's Theory itself, and we have also deleted the
reference to Defant (1958) from the table (now numbered Table 1) since it is now clearer in the text.
• **The altered text now reads:** "Table 1 summarizes these data, and contrasts them with those from Equilibrium Theory
(values obtained from Defant, 1958), while Table A1 catalogues the detailed results".
**95 -** *'data results' –> results*
**98 -** *reinforces –> shows*
**101 -** *.. amplitude (Figure 3c).*
**102** - *in the text and tables and figures it would be much simpler if you dropped the 'a' and have M2 for example to refer to*
*its amplitude. All the a's make things messy. You would have to say you were doing that of course.*
**103 -** *drop relatively The two bullets below. Could you mention them as determining Type 1 and Type 4.*
**109 -** *surely that is not referring to Figure 1, you can't see Cook Strait in that at all*
**110 -** *75 to 90% of what? What are the adjacent coasts?*
**114 -** *'anomalistic timeframes' –> 'a month' and drop the 27.5546 four decimals –> 27.6 will do*
• **Response:** All changes made exactly as suggested. Figure 1 has been corrected to Fig. 2. For 110, see new text below.
• **The altered text now reads:** "Type 1 regimes occur on the Kapiti and Cook Strait area (Fig. 2), where the $N_2$ and
$M_2$ amplitudes reduce by 75 to 90%, but the $S_2$ amplitude reduces by only about 30%, compared to on the western
coasts both north and south of this central NZ area".
**116** *Chatham Rise and Castle Point are not in Figure 2.*
• **Response:** The text read: "*This type of regime occurs, for example, around the northern Chatham Rise near Kaikoura,*
*and as far north as Castlepoint on the east coast of the South Island*". We have left this text and the Fig. 2 map
unchanged regards these labels as both Castlepoint and Kaikoura were on this figure (see north and south of Cook
Strait on the central east coast). Other labels that were missing from Fig. 2 as pointed out below (e.g. North/ South/
Stewart Island) have been now added though, and the Castlepoint dot has been shifted ~3 mm north, and the spelling
corrected from Castel Point to Castlepoint, in response to the Glen Rowe review comment (below).
**121** *sentence 'By examining'. I would drop this sentence. You repeat yourself a lot.*
• **Response:** Deleted – thank you, we really appreciated all of your suggestions for shortening our text and removing
repetition. The paper is now much tighter as a result.
**126 -** *I would say spring-neap and then perigean-apogean as that is the order else- where Two bullets. Can you mention them*
*as Type 2 and 3*
**133 -** *amplitudes being only*
• **Response:** All changes made exactly as suggested.
• **The altered text now reads:** "The variability in these two ratios means that, except where we find 'spring-neap' or
'perigean-apogean' monthly tidal envelope types, spring-neap tides do occur but the overall monthly envelope shape
is fundamentally altered (asymmetrically) due to the perigean-apogean influence".
**139 -** *Sumner not in Figure 2 Equations 1 - drop the a's (see above). Also drop the 'more stable' words. I guess you mean*
*similar locally? But the same situation would apply if the constituents varied a lot spatially. These are just simple algebraic*
*relationships at a particular position – they have nothing to do with spatial scale or 'stability'.*
**161** *Table 4 should be 3?*
• **Response:** For 139, Sumner is/was in Fig. 2, on the central east coast of the south island (just above the peninsula –
this is a gauge site for Christchurch city). For Eqs. 2, 2a and 2b (formerly 1, 1a and 1b), all 'a's have been deleted
and it has been specified in the text that these ratios refer to amplitudes. All references to stability have been dropped.
Re '161', both Tables 3 and 4 now seemed superfluous after re-arranging Table 1 and with removal of the

 experiments, so these two tables (3 and 4) have been deleted, as has all mention of them, shortening our paper nicely
 (thank you for those useful suggestions).
•   **The altered text now reads:**
"Thus, in a similar manner to van der Stok's (1897) method for calculating *daily* tidal form factors, a *monthly* tidal envelope
factor (*E*) may be calculated for semi-diurnal tidal regions, including that of NZ, according to:

$E = \frac{M_2 + N_2}{M_2 + S_2}$,                                                             (2)

where $M_2$, $N_2$ and $S_2$ refer to the constituent amplitudes. This equation can be further expressed as:

$E = \frac{1 + \frac{S_2}{M_2}x}{1 + \frac{S_2}{M_2}}$,       with $x = \frac{N_2}{S_2}$                                      (2a)

$E = \frac{1 + \frac{N_2}{M_2}}{1 + \frac{N_2}{M_2}y}$,       with $y = \frac{S_2}{N_2}$                                       (2b)".

*173 - it is strange to read of M2, which is the largest, moderating something smaller. I think this paragraph needs rewording.*
*Also I don't understand (i) and (ii). What are the 'annual' and 'subsequent' things? I guess the R in MTR is ratio? Not clear.*
*But note that MTR stands for Mean Tidal Range in usual tidal studies. Anyway I found these experiments at lines 180-204*
*somewhat unconvincing, although I do understand why you felt the need to inject some rigour into the choice of boundary*
*values between Types. But the experiments do not cover the whole space of possibilities for amplitude and phase lag of all*
*constituents concerned. The main thing to me is Figure 6 which shows nicely how Fsm varies with x and y. Why don't you then*
*just define the boundaries between Type 1 etc. in an ad hoc way, similar to the way as F is divided in an ad hoc way for*
*'semidiurnal' etc. After all, in the end all these form factors are just handy coarse descriptive subdivisions for the tide. Anyway*
*lines 180-204 need rewriting - see my comment at line 173 also. It is just not clear what you are doing.*
•   **Response:** Thank you for this simplifying comment. The "moderating influence of" has been replaced by "strong
influence of". The whole section on the experiments, including Figure A1, has been deleted. As suggested, simplified
ad hoc (and less repetitive) explanations for selecting the boundaries between the different *E* types has been used
instead as follows.
•   **The altered text now reads:**
"Below we explain how we set boundaries between the different E types around NZ, using our case study data, and as
summarised in Fig. 6.
Firstly, in any semi-diurnal tidal regime (F<0.25) anywhere in the world where the amplitude ratio $\frac{N_2}{S_2} < 1$, spring-neap cycles will feature clearly in the tidal height records. Thus, the boundary separating Types 1 and 2 from Types 3 and 4 occurs at $\frac{N_2}{S_2} =$

1. Type 1 and 2 areas of the NZ coast are characterized by relatively larger $S_2$ amplitudes (19-40 cm) than areas with stronger
perigean-apogean influences (2-18 cm) (Table 1). Secondly, tidal regimes with stronger spring-neap signals include places
where spring-neap cycles occur as consecutive fortnightly cycles of similar magnitude (Type 1 or 'spring-neap' type regimes),
and places where spring-neap signals dominate but with noticeable variability in the magnitudes of consecutive cycles due to
subordinate perigean-apogean influences (Type 2 or 'intermediate, spring-neap' regimes). In NZ the strongest spring-neap
influence occurs in the Cook Strait to Kapiti area, where harmonic analysis revealed an amplitude ratio of $\frac{N_2}{S_2} = 0.35$ and an E

value of 0.79 (Table 1). Examining the shapes of tidal height plots showed that Kapiti had the only completely spring-neap
dominated tidal envelope amongst the case study sites. Hence the boundary between Type 1 versus 2 was set as E = 0.8 for
NZ, just greater than that of Kapiti and below the next strongest spring-neap influenced site, Nelson, where E = 0.9 (Fig. 6).
Lastly, to set a boundary between 'perigean-apogean' and 'intermediate, perigean-apogean dominant' regimes (i.e. Types 3
versus 4), we again examined tidal height plots to determine a boundary value of E = 1.15, between the 'intermediate, perigean-
apogean dominated' type regime of Napier (E = 1.147) and the 'perigean-apogean' type regime of Kaikoura (E = 1.162) (Table
A1; Fig. 6)".

*203 - see below. mention the other red blob.*
•   **Response:** The following text has been added:

•   **The altered text now reads:** "As shown in Fig. 1c, such regimes unusual internationally, also occurring in limited
areas of the Cook Islands and northeast of Pitcairn Islands in the Southwest Pacific Ocean; in Alaska's Bristol Bay,
Canada's Hudson Bay and offshore of the North Carolina to Virginia coast in North America; on the north coast of
the Bahamas in Central America; and in the Gulf of Ob in Russia".

*217 - if you agree then drop 'theoretical experiments' here.*
•   **Response:** Yes – all mention of the experiments, and the experiments themselves, have been dropped completely
from the paper.

*220 - these three do not all operate at 'synodic anomalistic timescales'. Why not just '..three key constituents (M2, S2 and*
*N2).' At this point it occurred to me that a similar exercise could be conducted for areas of predominantly diurnal (but a bit*
*mixed) tides. Could you speculate in this Discussion which parts of the world could benefit that way?*
•   **Response:** The 3 key constituents change has been made. Using the FES2014 model for a similar exercise, we
explored predominantly diurnal tidal regimes and found a possible area, the Ross Sea, Antarctica, where there are
extremely weak (F>15), very weak (F>8) and weak (F>5) $M_2$ and $S_2$ amplitudes along with the areas of $M_2 > S_2$, $S_2 > M_2$,
$N_2 > S_2$ and $S_2 > N_2$ amplitudes. Thus, our approach to classifying monthly tidal patterns can be applied to the Ross Sea
diurnal tide area, but it is not as simple as the application in this paper to semi-diurnal NZ regimes, so we have left
this exercise for another paper. We are thankful for this suggestion though.
•   **The altered text now reads:** "The result is a widely applicable monthly tidal envelope factor, $E$, for classifying semi-
diurnal regimes based on the amplitudes and amplitude ratios of three key constituents: $M_2$, $S_2$, and $N_2$".

*230 - this isn't necessarily true. Figure 1c shows where S2 is small compared to M2. It doesn't necessarily follow that perigean*
*influences dominate.*
•   **Response:** We checked the results for all 'red blob' areas, and in all of these the $N_2$ is at least five times greater than
the $S_2$, hence we remain confident in this point. We did, however substitute the word 'paramount' with 'stronger' and
'minor with 'very weak'.
•   **The altered text now reads:** "Figure 1b illustrates the division of the semi-diurnal areas of the world's oceans into
those where spring-neap cycles are the main monthly tidal envelope influence versus those where the perigean-
apogean signal is stronger, while Fig. 1c illustrates areas of the world's oceans where spring-neap signals are very
weak compared to 'perigean-apogean' influences in the monthly tidal envelope".

*239 - what is 'low-frequency coastal flooding'?*
•   **Response:** This unhelpful descriptor has been deleted.
•   **The altered text now reads:** "We hope that our work inspires other efforts to study tidal height variations at
timescales greater than daily, work which could draw renewed attention to the fundamental role of tidal water levels
in shaping coastal environments, including in hazards such as coastal flooding".

*Table A1. Line 1 - you don't show tidal ranges, this will be confusing for most people. What you show on the last line are*
*ranges of amplitudes and phase lags in your data set. Also the 'ranges' shown are crazy for some as shown e.g. see 6-360 for*
*K1. But 360 degrees is the same as 0 degrees! Line 2 - values. Also the header should mention you show Types. Say if the*
*phase lags shown are in Greenwich Mean Time or local time? if Greenwich then they are usually denoted by G.*
•   **Response:** Corrections to caption and number values (360 –> 0) made exactly as suggested, and line 2 deleted. Phase
lag reference added and notation amended to $G$. Also the columns in this table have been re-ordered in line with the
suggested re-ordering of the Table 2 (now Table 1) columns.
•   **The altered caption now reads:** "Table A1. Monthly tidal envelope ($E$) types and values, daily form factors ($F$), and
data on the amplitude and phase lag (relative to Greenwich) values of 5 tidal harmonic constants at 27 sea level
stations around New Zealand".

*Figure A1. I don't understand the 'under conditions summarised in Table A1'. Surely all one needs to know is which stations*
*were used for these 3 examples.*

• **Response:** Since the experiments have been deleted, this figure has also now been removed.

*261 - doesn't matter much but Figure 2 looks like a simple coastline map to me that one could make with GMT or Matlab, so*
*where do the fancy 'map layers' come in? And with an undesirable national coordinate system to boot instead of lat/lon?*

• **Response:** The coordinate system and map outline were supplied by Mr Thyne as CorelDraw map layers (plural) –
they were not particularly complicated layers but this is just a software difference where CorelDraw is set up to
separate diagrams into component part layers. The coordinate system has been amended to internationally
understandable Lat/Long, and the Acknowledgement note has been simplified to 'outline map to avoid queries from
readers unfamiliar with CorelDraw.

• **The altered acknowledgement now reads:** "Thank you to …. John Thyne for supplying the Fig. 2 outline map".

• **The altered Fig. 2 now looks like:**

[Figure]

**Figure 2. Location of New Zealand sea level observation stations investigated in this research: circles indicate LINZ sites, rectangles**
**indicate NIWA sites; each site is colored according to monthly tidal envelope type. Offshore islands are not shown to scale (Raoul**
**and Chatham Islands).**

***Table 1. Line 1.*** *The word 'interval' in tides refers to the times of high tide since passage of the moon. What you are showing*
*here are not intervals but the periods of beating of the shown pairs of constituents. And personally I would abandon columns*
*3 and 4 - you are not writing a text book here - certainly drop column 3 (and in M2/S2 – drop 'axial'. M2/N2 - drop 'relative'.*
*line 3 'during the siderial month' –> during a month). And I would drop the Note which doesn't add anything.*
- **Response:** Fully taking on board your comments and acknowledging the level of the journal audience, we have
completely deleted Table 1 (and also Tables 3 and 4, as per your additional comments below) and re-ordered the
remaining table (previously Table 2, but now labelled Table 1).
***Table 2. I*** *would have a column 1 showing Type. And I would move Example Sites to be a column 2. First line of that:*
*Equilibrium Theory (no footnote and no Note – you have already mentioned Defant in the text).*
- **Response:** Table columns re-ordered and changes made as suggested (note – now this original Table 2 is called Table
1, as per the table deletion changes in the point above).
- **The altered Table 1 (formerly Table 2) now looks like:**

**Table 1. Comparison of tidal constituent amplitudes, amplitude ratios (including daily tidal form factor, $F$, and monthly tidal envelope factor, $E$) and ranges between the four distinct types of monthly tidal envelope ($E$ types) found in the 27 case study semi-diurnal tide regimes of New Zealand, and compared to Equilibrium Theory amplitude ratios**

| $E$ type | Example sites | Amplitude (cm) | | | | | Amplitude ratio | | | | | | | $F$ value range, description | $E$ value range, description |
|---|---|---|---|---|---|---|---|---|---|---|---|---|---|---|---|
| | | $M_2$ | $S_2$ | $N_2$ | $K_1$ | $O_1$ | $\frac{S_2}{M_2}$ | $\frac{N_2}{M_2}$ | $\frac{N_2}{S_2}$ | $\frac{S_2}{N_2}$ | $\frac{S_2+N_2}{M_2}$ | $\frac{K_1}{M_2}$ | $\frac{O_1}{M_2}$ | | |
| n/a | Equilibrium Theory | - | - | - | - | - | 0.47 | 0.19 | 0.41 | 2.44 | 0.66 | 0.584 | 0.415 | 0.68 mixed, mainly semi-diurnal | n/a |
| 1 | Kapiti | 55 | 26 | 9 | 2 | 2 | 0.47 | 0.16 | 0.35 | 2.89 | 0.64 | 0.04 | 0.04 | 0.05 semi-diurnal | 0.79 spring-neap |
| 2 | Nelson, Manukau, Taranaki, Onehunga, Westport, Charleston, Pusegur Point | 78 to 133 | 19 to 40 | 17 to 25 | 2 to 6 | 1 to 4 | 0.24 to 0.3 | 0.18 to 0.22 | 0.58 to 0.89 | 1.12 to 1.74 | 0.45 to 0.48 | 0.02 to 0.06 | 0.01 to 0.05 | 0.04 to 0.07 semi-diurnal | 0.90 to 0.98 intermediate, spring-neap dominant |
| 3 | North Cape, Boat Cove and Fishing Rock (Raoul Island), Dog Island, Auckland, Bluff, Lottin Point, Tauranga, Korotiti Bay, Moturiki, Green Island, Port Chalmers, Sumner, Gisborne, Napier | 50 to 112 | 4 to 18 | 10 to 22 | 2 to 8 | 1 to 4 | 0.06 to 0.2 | 0.2 to 0.23 | 1.07 to 3.5 | 0.29 to 0.94 | 0.28 to 0.43 | 0.02 to 0.06 | 0.01 to 0.06 | 0.05 to 0.14 semi-diurnal | 1.01 to 1.15 intermediate, perigean-apogean dominant |
| 4 | Kaikoura, Owenga, Castlepoint, Wellington | 48 to 65 | 2 to 3 | 10 to 14 | 2 to 4 | 2 to 4 | 0.04 to 0.05 | 0.21 to 0.22 | 4.67 to 5.50 | 0.18 to 0.21 | 0.25 to 0.27 | 0.04 to 0.06 | 0.04 to 0.06 | 0.08 to 0.12 semi-diurnal | 1.16 to 1.18 perigean-apogean |

*Table 3 - I guess this does no harm but it just repeats what has been given in the text. I would drop it.*
• **Response:** This table is now deleted – with clearer formatting of Table 2 (now Table 1) and this comment in mind
we found that it was now superfluous.
*Table 4 - I don't understand this table. It is tied up with mention of the experiments, see comments above. I would drop this*
*table as well.*
• **Response:** This table is now deleted, in line with removal of the experiments as commented on above.
*Figure 1 - just a suggestion but perhaps all panels could be made the same size. You have (a) large but that is for the normal*
*F which is not the subject of this paper and can be found in many text books. Also for this, and also for the other colour maps*
*in Figs 3,4 etc. could you have an arrow on the max colour as you have points on the maps with values which are in overflow.*
*As for Figure 1 (c), you should mention somewhere in the text where the other red blob is. Near Tahiti? line 4 of caption '....*
*monthly tidal envelope using criteria described in section 3.' Then for (c) see my comment for line 230.*
• **Response:** All 3 maps are now the same size. The overflow issue has been eliminated from this figure. The red blobs
in 1 c have now been described in the text more fully, as indicated in a point above. For Fig. 1c, see reply above. The
caption for 1c has also been adjusted.
• **The altered Fig. 1 and caption are now as follows:**

[Figure]

**Figure 1. (a) Global distribution of daily form factor (*F*) values, indicating daily tidal regime types (*F*<0.25: semi-diurnal; *F*>0.25 to**
**F<1.5 mixed-mainly semi-diurnal; *F*>1.5 to *F*<3: mixed-mainly diurnal; and *F*>3: diurnal, according to the classification of van der**
**Stok 1897, and Courtier 1938); (b) the world's semi-diurnal tidal areas (*F*<0.25) divided into those where spring-neap (green) versus**
**perigean-apogean (blue) signals are the main influence on the monthly tidal envelope; and (c) semi-diurnal tidal regimes (in red)**
**where the $S_2/M_2$ constituent amplitude ratio is <0.04 and the spring-neap tidal signals are very weak as compared to perigean-**
**apogean signals, derived from FES2014 tidal harmonic constants.**

*Figure 2 - please use conventional lat/lon and not a national coordinate system no-one else will understand. As mentioned*
*above there are places in the text (e.g. Stewart Is.) not shown. '&' –> 'and'. When Figure 2 is first mentioned in the text there*
*is no mention of the Type 1, 2 etc. So you have to returrn to this figure after you discuss Figures 6 and 7 and mention the Types*
*in Fig.2, and then please also use the same colours for the Types here as in Fig 6.*
• **Response:** All changes made as suggested. Please see new Fig. 2 above.
*Fig 3 - arrows needed on colour scales e.g. for the overflow top-left of 3(d). The contour annotation bottom right of 3(f) is*
*messy, please thin out the annotations. Also drop 'Unit' in 'Unit mm'. line 1 of caption - 'Amplitudes for'. Drop 'horizontal'.*
*Line 2 - drop 'derived and'. Drop 'database'. 'at a scale of' –> 'on a grid of'*
• **Response:** The scale of 3f has been changed to eliminate the overflow issue (we had previously mapped all the
constituents using the same color scale but have now decided to alter colour scales to suit each plot better). The
contours have also been thinned out as recommended. 'Unit' has been deleted from the scale bar. All caption changes
have been made as suggested.
• **The altered Fig. 3 and caption are now as follows:**

[Figure]

[Figure]

**Figure 3. Horizontal distribution of amplitudes for the (a) M$_2$, (b) S$_2$, (c) N$_2$, (d) K$_1$, and (e) O$_1$ tides around NZ, and (f) the resultant distribution of $F$, daily tidal form factor values, as calculated from the FES2014 tide model on a grid of 1°/16×1°/16. Note that the amplitude color scales vary between plots a and e.**

*Fig 4 - as mentioned I would drop the a's in the headers and captions. Arrows on colour scales. line 1 of caption - drop 'horizontal'. Line 2 - drop 'database'. 'at a scale of' –> 'on a grid of'*

*Fig 5 - drop a's*

- **Response:** The overflow area in Fig. 4c has been clearly delineated and labelled, and the suggested caption changes are made, including removal of 'a's.
- **The altered Fig. 4 and caption are now as follows:**

[Figure]

[Figure]

**Figure 4. Distributions of tidal constituent amplitude ratios around NZ for: (a) $\frac{S_2}{M_2}$; (b) $\frac{N_2}{M_2}$; (c) $\frac{N_2}{S_2}$ and (d) $\frac{S_2+N_2}{M_2}$; as calculated using**

**the FES2014 tide model on a grid of 1°/16×1°/16. Note that the amplitude color scales vary between plots a and d.**

*Fig 6 - this is actually a useful plot. Use another colour instead of pink which is too much like red. drop a's. Use same colours*
*for the Types as in Fig 2. Add dotted or dashed lines also for the Fsm boundary values chosen to define Types*
*1-4. Also what would be useful also would be to have values from FES2014 for the whole NZ coastline - that might be a fiddly*
*computing exercise but is obviously possible.*

•   **Response:** Color, dashed line boundary, and 'a' changes made as suggested. We decided not to make a FES2014
model based version of this diagram for the whole NZ coast due to our focus on observational data here (including
the Walters et al. data comparison) and also since we use the FES2014 data to do this task (albeit in map rather than
plot form), classifying the whole NZ coast into E type categories, in our newly expanded Figure 7.

•   **The altered Fig. 6 and caption are now as follows:**

[Figure]

**Figure 6. Plot of the relationship between the $\frac{N_2}{S_2}$ and $\frac{S_2}{M_2}$ amplitude ratios (y and x axes respectively) versus $E$ values (shown as plot**
**contours), with data points corresponding to New Zealand waters Type 1 sites (red dots); Type 2 sites (green dots); Type 3 sites (blue**
**dots); and Type 4 sites (yellow dots), all from Table A1; and tidal data representative of the greater Cook Strait area (grey crosses)**
**from Walters et al. (2010, Tables 1 and 3).**

*Fig 7 - overflow arrow. could roughly the same colours be used as for Fig 6 as far as possible? That has red-green-blue-pink*
*for types 1-4 whereas this has green-yellowred more or less (the blue is not used). line 2 - .. see Figure 5 for definitions and*
*examples of ..*
• **Response:** All colour and caption changes made as suggested. Figures 2, 5, 6 and 7 now have the same colours for
all *E* types.
• **The altered Fig. 7 and caption are now as follows:**

[Figure]

**Figure 7. Distribution of monthly tidal envelope factor (*E*) values (a); and types (b); in the waters around New Zealand, including**
**in the Cook Strait area between the two main islands (c); calculated using FES2014 data. In (b), *E* type 1 areas are shown in red;**
**type 2 in blue; type 3 in green; and type 4 in yellow. See Figure 5 for definitions and examples of monthly tidal envelope factor classes**
**and patterns.**

*Place names: There is some inconsistency in the place names used in the paper. For example, Cook Strait is referred to as 'Cook Strait' (line 73 and others) and 'Te Moanao-Raukawa Cook Strait' (line 70 and Figure 2). Lines 60 and 61 give both alternative names for the North Island and South Island. The English and MaÅ ri names are alternatives; it is not necessary to use both - choose one form and use it consistently.*

*The official name for Stewart Island is 'Stewart Island / Rakiura', not 'Rakiura or Stewart Island' as shown in line 61.*

*It is recommended that place names used are as shown in the NZ Place Names Gazetteer. Cook Strait is just 'Cook Strait' (not an official names but a recorded one), 'Castle Point' (line 117) is 'Castlepoint' (official). 'Aotearoa New Zealand' has been used for the name of the country (and abbreviated to ANZ) but until an Act of Parliament is passed the country is 'New Zealand'.*

- **Response**: The approach taken was: at first mention of a name we used use both Te Reo Maori and English names, thereafter referring to each place by which of these two is the most commonly used name today. This was combined with including both languages on the map. The reason for including both official written language names at first mention/ on the map was to recognise, with equivalence, both types of official language name.  This was also pragmatic, to try to give our paper some time-proofing, since it is not uncommon today for place names in our country to revert officially from their English to their Maori version. By including both, we thought our paper might withstand such changes and still be readable in the future. However in recognition of the direction to use only one form for each place name, and recognising that Copernicus has an international audience, we have selected one name for each place and used that consistently. On the other hand, the case of the Castlepoint typo (Castle Point on line 117) was an error and we have fixed this now.
- **Please see the new Fig. 2** above in the reply to the Woodworth review.

*Line 82: It would be helpful to point out that the results of the analysis of the records from the additional 33 locations are presented in Figure 6, and a sentence or two summarising those results would be appropriate.*

- **Response**: Yes, we have made this change as suggested.
- **The revised text now reads:** "The Cook Strait's tides were explored in detail by Walters et al. (2010): our Fig. 6 includes a re-analysis of their data using the E ratios. Note that the Cook Strait data includes 4 sites in the Type 1 category, as well as a number of Type 2 and Type 4 sites, and one Type 3 site, revealing this small Strait to be a concentrated area of monthly tidal envelope diversity".

*Line 83: What does 'reach the strongest' mean?*

- **Response**: Reach has been changed to "are"
- **The revised text now reads**: "*where spring-neap tides are the strongest in the country*".

*Line 93 and 94: The text here states that Figures 3 and 4 map the constituent amplitudes and ratios listed in Table 1, but surely the figures are derived from the FES2014 model as stated in the captions for Figures 3 and 4.*

- **Response**: Table 1 has now been deleted from our paper, according to the Woodworth review suggestion, so any confusion created by our reference to this table is also now now deleted.

*Line 117: Castlepoint is on the east coast of the North Island.*

- **Response**: Thank you for picking up this typo – "South" has been corrected to "North".

*Line 125: What does 'combined variability' mean? The rest of this sentence is difficult to follow - a diagram might help?*

- **Response**: We have deleted "combined" and altered this paragraph to make the meaning clearer.

• **The revised text now reads**: "We distinguished these two envelope types via the tides generated by variability in the
amplitude ratios of $\frac{S_2}{M_2}$ and $\frac{N_2}{M_2}$ (i.e. of the spring-neap cycle, and perigean-apogean cycle, forming tides, respectively).
In brief, the $\frac{S_2}{M_2}$ and $\frac{N_2}{S_2}$ amplitude ratios vary widely around NZ, with highest values in the west, lowest values in the
east, and intermediate values to the north and south (Fig. 4). By comparison, the $\frac{N_2}{M_2}$ amplitude ratios are relatively
stable and high, except in a relatively small area of Cook Strait to the Kapiti coast, where this ratio drops and thus
spring-neap cycles predominate (see 'spring-neap' Type 1 regimes above). The variability in these two ratios means
that, except where we find 'spring-neap' or 'perigean-apogean' monthly tidal envelopes types, spring-neap tides do
occur but the overall monthly envelope shape is fundamentally altered (asymmetrically) due to the perigean-apogean
influence".
*Figure 2: Castlepoint is shown out of its true position.*
• **Response**: Thank you – the map has been adjusted. See the adjusted position in Fig. 2, in the reply to Woodworth.

**3. Reply to: *Interactive comment by Anonymous Referee #2, received and published 5 February 2020**

*Reviewer introduction: The paper is basically acceptable, and Figures 1b, 1c, and 6 are useful. Most of the paper is devoted to trying to find the numerical delineations between spring-neap and perigean regimes, and that is a little tedious, as the boundaries are bound to be fuzzy and perhaps not applicable everywhere, even in purely semidiurnal regimes. (For example, the moderating role of K2, which likely causes variations throughout the year, isn't brought up. This, however, isn't fatal, since this whole exercise is merely to produce rough rules of thumb.) I didn't spot anything that is clearly in error, just minor issues, listed below. Some of these issues involve odd, almost off-the-cuff remarks in the introductory material rather than in the technical material. Numbers below refer to Line Numbers in the paper.*

- **Response**: Thank you for taking the time to review our paper. We feel we have been able to very significantly improve our paper based on the 3 sets of feedback provided. Please see below responses to the individual comments relevant to this review.

*24 - Neither the Egbert nor Stammer papers have anything to do with sea level change or gravimetry.*

- **Response**: The Egbert et al. (1994) reference has been deleted from here (details in our response to the Woodworth review). Stammer et al. (2014, p243) state as a justification for their accuracy assessment paper that "*An especially important application for accurate tide models is providing tide "corrections" to various measurements so that smaller nontidal signals may be studied. For example, barotropic tide models are used regularly to remove tidal variability from space geodetic observations; this is a critical necessity for successful satellite altimetry [e.g., Fu and Cazenave, 2001] and satellite gravimetry [Seeber, 2003; Visser et al., 2010], and in both cases improved tidal corrections lead to a reduction of aliased tidal "noise" in nontidal signals of interest*". It is this point from Stammer et al. (2014) that we wished to point our readers to. Our revised text now makes this clearer by the repositioning of this reference.
- **The revised text now reads**: "An understanding of tidal water level variations is fundamental to…. accurately resolving non-tidal signals of global interest (Stammer et al., 2014), such as in studies of sea level change".

*41 (also Table 1): Is it a sidereal month or a tropical month?*

- **Response**: This table has been deleted (see response to Woodworth review).

*47: "Far less attention" - There is a good reason for that, as the major tides are obviously most important for prediction. And why specify "modern" in this context? It's always been the case.*

- **Response**: Yes, thank you for your comment – we have removed this text and recalibrated the tone of remaining text.
- **The revised text now reads:** "Tidal envelopes at monthly scales depend on tidal regime. In general, semi-diurnal tidal regimes often feature two spring-neap tidal cycles per synodic (lunar) month. These two spring-neap tidal cycles are usually of unequal magnitude, due to the effect of the moon's perigee and apogee, which cycle over the period of the anomalistic month. In contrast, diurnal tidal regimes exhibit two pseudo spring-neap tides per sidereal month. For semi-diurnal regions where the $N_2$ constituent contributes significantly to tidal ranges, tidal envelope classification should consider relationships between the $M_2$, $S_2$, and $N_2$ amplitudes. The waters around NZ represent one such region: here the daily tidal form is consistently semi-diurnal, but large differences occur between sites within this region in terms of their typical tidal envelope types over fortnightly to monthly timescales. More than eighty years after the development of the ever-useful daily tidal form factors, attention to the regional distinction between different tidal envelope types within the semi-diurnal category forms the motivation for this paper".

*216 "having common ways of describing different types of tidal envelope is essential for living safely and productively..." – ESSENTIAL, really? That seems overblown. In fact, I consider a full-up tide prediction to be far more essential. Along the same lines, is it really necessary to have similar statements in the Abstract? The first and last sentences of the Abstract seem to me to be quite a stretch in trying to justify the work.*

•  **Response**: We have deleted 'essential' from our text here and replaced it with 'helpful'. The term in our revised
abstract reads 'of use', which is a much milder claim than previously written. The abstract has been much modified
in response to a similar point made in the Woodworth review above for the revised text.
Please refer to the response to the Woodworth review above for the revised text.

*60: what plates NZ sits on is rather irrelevant to the subject.*
•  **Response**: The names of the plates are not essential to our core paper topic but are useful in the context of explaining
the long narrow shape of the chain of islands that make up New Zealand, and this shape plays a role in interacting
with our ocean tides.
•  **The revised text now reads:** "New Zealand (Fig. 2) is a long (1600 km), narrow ($\leq$400 km) country situated in the
south-western Pacific Ocean and straddling the boundary between the Indo-Australian and Pacific plates. Its three
main islands, the North Island, the South Island, and Stewart Island/ Rakiura, span a latitudinal range from about 34°
to 47° South".

*88: I'm not sure why "sidereal" is used in reference to K1 and O1. "Declinational" or just "diurnal" seems more apt.*
•  **Response**: Yes thank you - we have replaced 'sidereal' with 'diurnal' in our revised text.

*173: "moderating" is an odd way to refer to M2. Table A1. It should state these are Greenwich phase lags (which I believe to*
*be the case), since lower-case "g" is often used to denote a local phase. One could also argue that the F value based on*
*"Equilibrium Theory" ought to be a function of latitude.*
•  **Response**: Thank you for these comments – we have addressed all of them. Table A1 now has correct reference to
Greenwich phase lags in the caption and the corrected capital G parameter label in the table proper. Regards line 173,
we removed the sentence with "moderating" in it (see details in the reply to the Woodworth review).
•  **The revised text now reads:** "We distinguished these two envelope types via the tides generated by variability in the
amplitude ratios of $\frac{S_2}{M_2}$ and $\frac{N_2}{M_2}$ (i.e. of the spring-neap cycle, and perigean-apogean cycle, forming tides, respectively).
In brief, the $\frac{S_2}{M_2}$ and $\frac{N_2}{S_2}$ amplitude ratios vary widely around NZ, with highest values in the west, lowest values in the
east, and intermediate values to the north and south (Fig. 4)".

*Is Table 4 really necessary? Aren't Tables 2 and 3 and Figure 6 sufficient?*
•  **Response**: Thank you and yes, Table 4 was unnecessary, so we have deleted this table (see response to Woodworth
review where we expand on our table deletions and adjustments). Basically only a revised version of the original
Table 2 remains (now re-labelled Table 1), with the original Tables 1, 3 and 4 deleted.

*And finally a point on names. Presumably the government of New Zealand has not (yet?) changed the country name to*
*Aotearoa. Is there a reason to use (what I assume is) Maori throughout this paper – including even for the Pacific Ocean and*
*Tasman Sea? I suspect that indigenous Australians have a different name for these. Why not use those? Why not use Korean*
*as well? I don't really see the point of using an obscure indigenous name for the Pacific Ocean.*
•  **Response**: Recognising that Copernicus has an international audience we have selected only one name for each place
within the country now, called the country 'New Zealand', and used English ocean names consistently throughout
the paper and in a revised version of Figure 2 (also see response to the Rowe comment for more details, and the
Woodworth review response for the new Fig. 2).

[revised manuscript text omitted]

Figure 5. Idealized examples of four different monthly tidal envelopes over one year, calculated using the amplitude value $M_2 = 100\ cm$ and the amplitude ratio values of: (a) $\frac{S_2}{M_2} = 0.46$, $\frac{S_2}{N_2} = 11.5$, $\frac{N_2}{M_2} = 0.04$; (b) $\frac{S_2}{M_2} = 0.27$, $\frac{S_2}{N_2} = 1.5$, $\frac{N_2}{M_2} = 0.18$; (c) $\frac{S_2}{M_2} = 0.12$, $\frac{S_2}{N_2} = 0.5455$, $\frac{N_2}{M_2} = 0.22$; and (d) $\frac{S_2}{M_2} = 0.04$, $\frac{S_2}{N_2} = 0.1818$, $\frac{N_2}{M_2} = 0.22$. Note that the $E$ values of these plots are: (a) 0.71; (b) 0.93; (c) 1.09; and (d) 1.17.

[Figure]

**Figure 6. Plot of the relationship between the $\frac{N_2}{S_2}$ and $\frac{S_2}{M_2}$ amplitude ratios (y and x axes respectively) versus $E$ values (shown as plot**
**contours), with data points corresponding to New Zealand waters Type 1 sites (red dots); Type 2 sites (green dots); Type 3 sites (blue**
**dots); and Type 4 sites (yellow dots), all from Table A1; and tidal data representative of the greater Cook Strait area (grey crosses)**
**from Walters et al. (2010, Tables 1 and 3).**

[Figure]

**Figure 7. Distribution of monthly tidal envelope factor (E) values (a); and types (b); in the waters around New Zealand, including**
**in the Cook Strait area between the two main islands (c), calculated using FES2014 data. In (b), E type 1 areas are shown in red;**
**type 2 in blue; type 3 in green; and type 4 in yellow. See Figure 5 for definitions and examples of monthly tidal envelope factor classes**
**and patterns.**

---

## Author Response (AR2)

*Response to final review of the paper* **"A monthly tidal envelope classification**
**approach for semi-diurnal regimes with variability in $S_2$ and $N_2$ tidal amplitude**
**ratios"** *by Philip Woodworth, submitted 21 April 2020*

Do-Seong Byun[1], Deirdre E. Hart[2]
[1]Ocean Research Division, Korea Hydrographic and Oceanographic Agency, Busan 49111, Republic of Korea
[2]School of Earth and Environment, University of Canterbury, Christchurch 8140, Aotearoa New Zealand
*Correspondence to*: Deirdre E. Hart (deirdre.hart@canterbury.ac.nz)

**Introduction** We are grateful for the final Woodworth review on this paper (received via the Editor's letter) as it has been
useful in helping with a final polish and improvement of the paper. Below we have copied each individual reviewer comment,
written below it a response in blue font, and then copied any insertions or modifications to the text. Almost all suggested
changes have been adopted wholesale, with the exception of redefining the monthly tidal envelope types based on one instead
of two ratios (please see below where we explain that we added your idea but also kept the two ratios). Following our responses
and changes made sections, we include in this file a revised version of the full paper with track changes. We will submit the
revised final paper file separately as well, according to the Copernicus instructions.
*Comments on resubmission of 'A monthly tidal envelope classification approach for semi-diurnal regimes with variability in*
*S2 and N2 tidal amplitude ratios' by Byun and Hart (OS special issue)*
*I didn't look back in great detail on my comments on the first version and the replies of the authors, I decided to just read it*
*afresh. I think it now reads much better than it did before, although it is still a bit repetitive.*
*I list below some things that I noticed with the new version but couple of main things:*
• **Response:** Thank you for this additional, careful review.
*One thing that looks wrong is sentence lines 126-128. This sentence is ok for S2/M2 but the ratio variation is the opposite for*
*N2/S2. Just look at Figure 4(a,c). Rewording needed.*
• **Response:** Thank you for pointing this out: we have fixed the wording of this sentence.
• **The altered text now reads:** "We distinguished these two envelope types via the tides generated by variability in the
amplitude ratios of $\frac{S_2}{M_2}$ and $\frac{N_2}{M_2}$ (i.e. of the spring-neap cycle, and perigean-apogean cycle, forming tides, respectively).
In brief, the $\frac{S_2}{M_2}$ amplitude ratio varies widely around NZ, with highest values in the west, lowest values in the east,
and intermediate values to the north and south, while variation in the $\frac{N_2}{S_2}$ amplitude ratio exhibits an opposite pattern
(compare Fig. 4a to 4c).
*The other is that, as I mentioned last time, the most interesting figure to me is the present Figure 7 which made me wonder at*
*the value of the new E. Forget Type 1 for the moment which has only one red dot. You can see that using E to decide between*
*types 3 and 4 at 1.15 is problematical - the curve just scrapes the blue and yellow dots. A more efficient selection would be*
*simply by N2/S2 ratio at 4.0, as for the division between types 2 and 3 at 1.0. I hope you see what I mean and can put in a*
*sentence along these lines in the Discussion.*
• **Response:** We understand you mean Figure 6 (the E type separating plot that you identified as most interesting in
your first review). Thank you for this suggestion. Upon careful consideration and understanding of your idea, we are
still using both ratios in defining our boundaries but we also accommodated your point in new section 3.2 text (see
yellow highlight below) that spells out the boundaries including comment regarding how our Type 3 and Type 4 dots
are separated by distinct N2/S2 ratios boundaries (though so too are the two type 3 clusters – read on). We have also
minimised the size of the dots to improve the appearance of dots skimming the boundary lines, but cannot make the
dots too small or their colour distinction disappears. We put back a third decimal place in the E ratio data in Table 1
and A1 as we realised, thanks to your comment, that rounding to 2 decimal places obscured the actual point locations
relative to the boundary lines.

As you see in Fig. 7, the distinction between types 2 and 3 (green and blue dots) at N2/S2 = 1.0 is as close to the line
as the Type 3 and 4 distinction that you highlight as a problem (yellow and blue dots either side of the E=1.15 line).
Despite the proximity of dots to the N2/S2 = 1.0, we are confident that this is a proper boundary since it denotes
where perigean/apogean influences dominate over spring/neap influences, and vice versa. There is some separation
between the green Type 2 cluster and start of the blue Type 3 cluster on the S2/M2 x-axis, making this axis relevant
to this boundary.
As you point out, there is a y-axis gap (around N2/S2 = 4) between the yellow and blue dots. But there is another gap
on the y-axis, in the middle of the blue dots, at around N2/S2=2. When we plot these monthly tidal envelope types
out, the blue dot sites all have mixed perigean/apogean and spring/neap influences, but all with perigean/apogean
being the dominant influence.
We therefore feel that the combination of the two ratios is a better way to separate out the types overall, compared to
using the N2/S2 alone, acknowledging the gaps on both ratio axes.

• **The altered text now reads:**

"The boundaries between our different NZ monthly tidal envelope types were as follows:

-      63     $E < 0.8$ indicates a Type 1 'spring-neap' regime;
-      64     $E$ between 0.8 and 1.0 indicates a Type 2 'intermediate, predominantly spring-neap' regime (with the upper
bound also corresponding to an amplitude ratio of $\frac{N_2}{S_2} < 1$ in semidiurnal regimes);
-      66     $E$ between 1.0 and 1.15 indicates a Type 3 'intermediate, predominantly perigean-apogean' regime (with the
lower bound also corresponding to an amplitude ratio of $\frac{N_2}{S_2} > 1$ in semidiurnal regimes); and
-      68     $E > 1.5$ indicates a Type 4 'perigean-apogean' regime (with the lower bound also corresponding to an amplitude
ratio of $\frac{N_2}{S_2} > 4$ in our NZ regimes)".

• **The altered Fig. 6 now looks like:**

[Figure]

**Figure 6. Plot of the relationship between the $\frac{N_2}{S_2}$ and $\frac{S_2}{M_2}$ amplitude ratios (y and x axes respectively) versus $E$ values (shown as plot**
**contours), with data points corresponding to New Zealand waters Type 1 sites (red dots); Type 2 sites (green dots); Type 3 sites (blue**
**dots); and Type 4 sites (yellow dots), all from Table A1; and tidal data representative of the greater Cook Strait area (grey crosses)**
**from Walters et al. (2010, Tables 1 and 3).**

*Other things in line order:*
*title - I wondered if the title would be better as 'A monthly tidal envelope classification for semi-diurnal tidal regimes in terms*
*of the relative proportions of the S2, N2 and M2 constituents'. Or something like that, it is your paper so up to you.*
• **Response:** We have altered the title as suggested.
• **The altered text now reads:** "A monthly tidal envelope classification for semi-diurnal regimes in terms of the relative
proportions of $S_2$, $N_2$ and $M_2$ constituents".
*7-8 and 187-189 - I didn't understand 'access .. wharves'. What are you saying? That access to marine environments by boat*
*is difficult and/or to infrastructure? Needs rewording.*
• **Response:** Yes, we meant that the tide controls our access to marine environments when we need to use fixed
infrastructure like jetties to access the water by boat. It was a bit wordy so we simplified this text.
• **These two altered sections of text now read:**
"Daily tidal water level variations are a key control on shore ecology; on access to marine environments via ports,
jetties and wharves; on drainage links between the ocean and coastal hydrosystems such as lagoons and estuaries; and
on the duration and frequency of opportunities to access the intertidal zone for recreation and food harvesting
purposes".
"The daily water level variations wrought by the tides are a key control on shore ecology and on the accessibility of
marine environments via fixed port, jetty and wharf infrastructure. These variations also moderate the functioning of
drainage links between the ocean and coastal hydrosystems; and determine the duration and frequency of
opportunities to access the intertidal zone for recreation and food harvesting purposes".
*26-27 - as I mentioned before these references are all about tides rather than non-tidal signals. But ok. But I would add*
*Stammer et al to the others - there is no reason to separate it.*
• **Response:** We have placed all the references together at the end of this sentence again.
• **The altered text now reads:** "An understanding of tidal water level variations is fundamental to resilient inundation
management and coastal development practices in such places, as well as to accurately resolving non-tidal signals of
global interest such as in studies of sea level change (Cartwright, 1999; Masselink et al., 2014; Olson, 2012; Pugh,
1996, Stammer et al., 2014)".
*137 - Fig 3 and 4*
• **Response:** This change has been made as suggested.
• **The altered text now reads:** "…(Fig. 3 and 4…".
*142 .. ratio are between ..*
• **Response:** This change has been made.
• **The altered text now reads:** "Here values of the $\frac{N_2}{S_2}$ amplitude ratio are between 1.07 and 3.5, while values of the
$\frac{S_2+N_2}{M_2}$ amplitude ratio are between 0.28 and 0.43 (Fig. 4, Table 1)".
*146 - you have UK spelling here (colour) and US (color) in the figure captions. I suggest you use UK spelling throughout as*
*Copernicus is a European journal.*
• **Response:** The spelling has been changed to UK throughout. Color, analyzed and characterized have been replaced
with colour, analysed and characterised, in particular.
*162 - please add '.. occurs at N2/S2 =1 when also E = 1.'*
• **Response:** Change made as suggested.
• **The altered text now reads:** "Thus, the boundary separating Types 1 and 2 from Types 3 and 4 occurs at $\frac{N_2}{S_2} = 1$,
when also $E = 1$".

*You need to spell out what the divisions between types are.*
• **Response:** Thank you for this prompt. Yes, we have added a clear statement spelling out the boundaries in list form
as follows.
• **The altered text now reads:**
"The boundaries between our different NZ monthly tidal envelope types in NZ waters were as follows:
• $E < 0.8$ indicates a Type 1 'spring-neap' type regime;
• $E$ between 0.8 and 1.0 indicates a Type 2 'intermediate, predominantly spring-neap' type regime (with the upper
bound also corresponding to an amplitude ratio of $\frac{N_2}{S_2} < 1$ in semidiurnal regimes);
• $E$ between 1.0 and 1.15 indicates a Type 3 'intermediate, predominantly perigean-apogean' type regime (with
the lower bound also corresponding to an amplitude ratio of $\frac{N_2}{S_2} > 1$ in semidiurnal regimes); and
• $E > 1.5$ indicates a Type 4 'perigean-apogean' type regime (with the lower bound also corresponding to an
amplitude ratio of $\frac{N_2}{S_2} > 4$ in our NZ regimes)".
*This is also a point to mention that you call these things Types here, but in the tables you call them E-Types. Because I think*
*they can just as easily be defined by N2/S2 (see the second of my main points above) I think they would be better as Types*
*throughout.*
• **Response:** We have changed all table and text mentions of "E types" to read "envelope types" for consistency.
• **The altered text now reads:** "envelope types".
*lines 181-185. There are several problems here:*
*- one is the last sentence returns to mention of NZ after you have digressed to the rest of the world. Could you somehow work*
*the last sentence after line 180 probably, then go onto other places.*
• **Response:** We have reworded this paragraph to position the non-NZ places after all of the NZ place descriptions. We
have also reworded the list of international (non-NZ) places and deleted reference to 'North America'.
• **The altered text now reads:** "In summary, Fig. 7 illustrates the monthly tidal envelope values and types in the waters
around NZ using $E$. The west coast is characterised by Type 2 monthly tidal envelopes, with two unequal spring-neap
cycles per month. As mentioned above, Type 1 monthly tidal envelopes, with their defined spring-neap tides, are only
found in the western Cook Strait to Kapiti coast area. The Cook Strait's tides were explored in detail by Walters et
al. (2010): our Fig. 6 includes a re-analysis of their data using the $E$ ratios. Note that the Cook Strait data includes 4
sites in the Type 1 category, as well as a number of Type 2 and Type 4 sites, and one Type 3 site, revealing this small
Strait to be a concentrated area of monthly tidal envelope diversity. Extensive areas of Type 3 'intermediate, perigean-
apogean dominated' monthly tidal envelope are found along the northeast and southeast coasts of NZ, while the
central east coasts show Type 4 'perigean-apogean' tidal envelopes. As shown in Fig. 1c, such regimes are unusual
internationally, also occurring in limited areas of the Cook Islands; northeast of the Pitcairn Islands; in Canada's
Hudson Bay; in Alaska's Bristol Bay; offshore of the North Carolina to Virginia coast in the Unites States of America;
on the north coast of the Bahamas; and in the Gulf of Ob in Russia".
*Another problem with your lists in this paper is you seem to use semicolons and commas randomly. In lists one starts with a*
*colon, with semicolons between items (or some people have commas). Please can you go through and tidy that up?*
• **Response:** Yes – the reason why it seemed that we mixed commas and semi-colons in the above list of places was
that there was a regional groupings separated by semi-colons with countries separated by commas within the regional
groups. However we simplified this, as above, to remove the regional groupings and just use semicolons. We have
checked the lists throughout the rest of the paper text and use semicolons throughout now, except in the Fig. 6 caption
alone, where there are types separated by commas and data groupings separated by semicolons. We hope that you
can suggest if you would prefer some other pattern to this one figure caption punctuation during the proofing stage.

| | |
|---|---|
| 172 | *- then a couple of these examples such as N Carolina/Virginia or Gulf of Ob just don't appear on Fig 1c, no doubt because the* |
| 173 | *dots are so small. So you should say that to save people wasting time looking for them. Where is the Gulf of Ob anyway?* |

*- then a couple of these examples such as N Carolina/Virginia or Gulf of Ob just don't appear on Fig 1c, no doubt because the*
*dots are so small. So you should say that to save people wasting time looking for them. Where is the Gulf of Ob anyway?*
• **Response:** As stated in the text above, the Gulf of Ob is in Russia. All of the places mentioned are marked by red
blobs in Fig. 1c though some are indeed small – they should be readily visible if people view the electronic version
of the paper figure but perhaps not easily seen on a printed copy if this figure is reproduced at column instead of page
width. If possible we recommend that this Fig.1 is reproduced at page width.
*187-189 copies verbatim from the abstract. Could you maybe change the wording a little to show willing?*
• **Response:** Yes – we have altered this text so it is now different, not verbatim. Please see the new text for both section
in response to the comment on lines 7-8 and 187 to 189 earlier in this document.
*228 - reword something like: Monthly tidal envelope types and values of monthly (E) and daily (F) form factors ... of 5 tidal*
*constituents ...*
• **Response:** This has been reworded as suggested.
• **The altered text now reads:** "Table A1. Monthly tidal envelope types and values of monthly ($E$) and daily ($F$) form
factors, and data on the amplitude ($a_i$) and phase lag ($G_i$, relative to Greenwich) values of 5 tidal constituents'
(subscript $i$) harmonic constants at 27 sea level stations around New Zealand".
*column 2 - Type*
• **Response:** This has been modified to envelope type instead of $E$ type.
• **The altered text now reads:** "Envelope type".
*As I mentioned before, in the last line you don't seem to realise that 0 and 360 deg is the same thing. This range is not a very*
*useful anyway when you have shown there are amphidromes all over the place. I would either not include that whole line or*
*just show the ranges of amplitude.*
• **Response:** We have deleted the phase lag ranges from this table, as recommended.
*279 - Chichester*
• **Response:** This typo has been fixed.
• **The altered text now reads:** "Chichester".
*Table 1, line 300 - E types --> types*
• **Response:** This has been modified to envelope types instead of $E$ types.
• **The altered text now reads:** "the four distinct monthly tidal envelope types found in the 27 case study semi-diurnal
tide regimes of New Zealand".
*column 1 - Type*
• **Response:** This has been modified to eliminate "E" type.
• **The altered text now reads:** "Envelope type".
*Figure 2 is fine but it will probably be printed at half this size so I would make all the fonts larger. Also when it is smaller you*
*won't be able to tell circles from rectangles so they should be larger also. colored - see above*
• **Response:** We have redrawn this figure with larger font and larger dots as recommended. We realised, thanks to your
comment, that the circles and rectangles differentiating the NIWA and LINZ data sources was an unnecessary
complication, so we changed them all to be the same circular form and have deleted reference to these two different
data providers. We also changed the language to UK English.
• **The altered Fig. 2 now looks like:**

[Figure]

**Figure 2. Location of New Zealand sea level observation stations investigated in this research. Each site is coloured according to monthly tidal envelope type. Offshore islands are not shown to scale (Raoul and Chatham Islands).**

*Fig 3 caption - drop Horizontal*
• **Response:** This has been deleted.
• **The altered text now reads:** "Figure 3. Distribution of amplitudes for the (a) M2, (b) S2, (c) N2, (d) K1, and (e) O1
tides around NZ, and (f) the resultant distribution of F, daily tidal form factor values, as calculated from the FES2014
tide model on a grid of 1°/16×1°/16. Note that the amplitude colour scales vary between plots a and e".

*Fig 5 - it doesn't matter much but looks a bit odd to show one ratio like 0.46 to 2 decimals and one like S2/N2 to four*
• **Response:** Yes, we have rounded the two 4 decimal place numbers to 2 decimal place numbers for consistency.
• **The altered caption now reads:** "Figure 5. Idealized examples of four different monthly tidal envelopes over one
year, calculated using the amplitude value $M_2 = 100$ cm and the amplitude ratio values of: (a) $\frac{S_2}{M_2} = 0.46, \frac{S_2}{N_2} = 11.5,$
$\frac{N_2}{M_2} = 0.04$; (b) $\frac{S_2}{M_2} = 0.27, \frac{S_2}{N_2} = 1.5, \frac{N_2}{M_2} = 0.18$; (c) $\frac{S_2}{M_2} = 0.12, \frac{S_2}{N_2} = 0.54, \frac{N_2}{M_2} = 0.22$; and (d) $\frac{S_2}{M_2} = 0.04,$
$\frac{S_2}{N_2} = 0.18, \frac{N_2}{M_2} = 0.22$. Note that the *E* values of these plots are: (a) 0.71; (b) 0.93; (c) 1.09; and (d) 1.17".

*Fig 6, line 1 - versus E values --> and E values. It is not versus which is when you have y vs. x.*

*I looked at this draft on paper and I couldn't see the grey points at all - I could on the pdf - so I would make them a bit darker.*
*Also they are stars (or asterisks) and not crosses.*
• **Response:** We have changed 'versus' to 'and'. We have re-drawn the figure with the 'grey crosses' (which, yes, were
actually stars) as black stars and fixed the caption to reflect this.
• **The altered caption now reads:** "Figure 6. Plot of the relationship between the $\frac{N_2}{S_2}$ and $\frac{S_2}{M_2}$ amplitude ratios (y and x
axes respectively) and $E$ values (shown as plot contours), with data points corresponding to New Zealand waters
monthly tidal envelope Type 1 sites (red dots); Type 2 sites (green dots); Type 3 sites (blue dots); and Type 4 sites
(yellow dots), all from Table A1; and tidal data representative of the greater Cook Strait area (black stars) from
Walters et al. (2010, Tables 1 and 3)".

*Fig 7 caption - (a) Distribution ... values and (b)tidal types in .. including (c) in the . islands, all calculated using .. In (b) and*
*(c) ..*
• **Response:** Changes made exactly as recommended.

[revised manuscript text omitted]